# Learning from Time-dependent Streaming Data with Online Stochastic Algorithms

**Antoine Godichon-Baggioni**                                   *antoine.godichon__baggioni@sorbonne-universite.fr*
*Laboratoire de Probabilités, Statistique et Modélisation*
*Sorbonne Université*

**Nicklas Werge**                                              *nicklas.werge@sorbonne-universite.fr*
*Laboratoire de Probabilités, Statistique et Modélisation*
*Sorbonne Université*

**Olivier Wintenberger**                                       *olivier.wintenberger@sorbonne-universite.fr*
*Laboratoire de Probabilités, Statistique et Modélisation*
*Sorbonne Université*

**Reviewed on OpenReview:** *https://openreview.net/forum?id=kdfiEu1ul6*

## Abstract

This paper addresses stochastic optimization in a streaming setting with time-dependent and biased gradient estimates. We analyze several first-order methods, including Stochastic Gradient Descent (SGD), mini-batch SGD, and time-varying mini-batch SGD, along with their Polyak-Ruppert averages. Our non-asymptotic analysis establishes novel heuristics that link dependence, biases, and convexity levels, enabling accelerated convergence. Specifically, our findings demonstrate that (i) time-varying mini-batch SGD methods have the capability to break long- and short-range dependence structures, (ii) biased SGD methods can achieve comparable performance to their unbiased counterparts, and (iii) incorporating Polyak-Ruppert averaging can accelerate the convergence of the stochastic optimization algorithms. To validate our theoretical findings, we conduct a series of experiments using both simulated and real-life time-dependent data.

## 1 Introduction

Machine learning has experienced remarkable growth and adoption across diverse domains, revolutionizing various applications and unleashing the potential of data-driven decision-making (Bishop & Nasrabadi, 2006; Goodfellow et al., 2016; Sutton & Barto, 2018; Hastie et al., 2009; Hazan et al., 2016; Shalev-Shwartz et al., 2012). With this proliferation of machine learning, a significant volume of new data has emerged, including streaming data that flows continuously and poses unique challenges for learning algorithms. Unlike traditional static datasets, streaming data requires algorithms to adapt in real-time, continuously updating their models to accurately predict new samples.

At the heart of machine learning lies optimization, the process of finding optimal model parameters that minimize the objective function and extract meaningful insights from the data (Abu-Mostafa et al., 2012). Stochastic Optimization (SO) methods have emerged as powerful tools for addressing optimization tasks in machine learning (Kushner & Yin, 2003; Nemirovskij & Yudin, 1983; Shalev-Shwartz & Ben-David, 2014). These SO methods overcome the scalability limitations of traditional batch learning techniques and allow for efficient processing of streaming data (Bottou et al., 2018).

Among the various SO methods, Stochastic Gradient Descent (SGD) and its variants stand out as the workhorses of the field (Robbins & Monro, 1951), extensively utilized in machine learning tasks (Hardt et al., 2016; Shalev-Shwartz et al., 2011; Zhang, 2004; Xiao, 2009). SGD and its variants provide a practical and

efficient approach to optimize complex models by updating the model parameters using stochastic estimates of the gradients. These methods have proven their effectiveness in a wide range of applications, including deep learning, natural language processing, and computer vision (Goodfellow et al., 2016; Sutton & Barto, 2018; Hastie et al., 2009).

However, traditional analyses of SO problems often assume that gradient estimates are unbiased and drawn independently and identically distributed (i.i.d.) from an unknown data generation process (Cesa-Bianchi et al., 2004). Unfortunately, real-world data rarely adheres to this idealized assumption. Streaming data, in particular, introduces additional complexity, as dependencies and biases can arise due to time-dependency or other factors inherent in the data generation process. These dependencies and biases pose significant challenges for conventional SO methods.

Notably, SGD-based methods have demonstrated the ability to converge even under biased gradient estimates (Ajalloeian & Stich, 2020; Bertsekas, 2016; d'Aspremont, 2008; Devolder et al., 2011; Gorbunov et al., 2020a;b; Schmidt et al., 2011). However, the theoretical understanding of the convergence properties of SGD in the presence of biased gradients is not yet well-established. While empirical studies have shown promising results in specific applications (Agarwal & Duchi, 2012; Chen & Luss, 2018; Karimi et al., 2019; Ma et al., 2022; Schmidt et al., 2011), more rigorous investigations are necessary to generalize these findings and comprehend the underlying mechanisms.

**Contributions.** In this paper, we explore SGD-based methods in the context of streaming data. We extend the analysis of the unbiased i.i.d. case by Godichon-Baggioni et al. (2023) to include time-dependency and biasedness. By leveraging their insights, we investigate the effectiveness of first-order SO methods in a streaming setting, where the assumption of unbiased i.i.d. samples no longer holds. Our non-asymptotic analysis establishes novel heuristics that bridge the gap between dependence, biases, and the convexity levels of the SO problem. These heuristics enable accelerated convergence in complex problems, offering promising opportunities for efficient optimization in streaming settings. Our contributions can be summarized as follows:

- **Non-asymptotic Convergence Rates of Time-varying Mini-batch SGD-based methods.** We present non-asymptotic convergence rates specifically tailored for time-varying mini-batch SGD-based methods under time-dependency and biasedness. These convergence rates offer valuable insights into achieving and enhancing convergence in applications involving time-dependent and biased inputs, providing a comprehensive understanding of the optimization process in streaming settings.

- **Breaking Long- and Short-term Dependence.** Our study demonstrates the effectiveness of SGD-based methods in overcoming long- and short-range dependence by leveraging time-varying mini-batches. These mini-batches are carefully designed to counteract the inherent dependency structures present in streaming data, leading to improved performance and convergence. This contribution expands the applicability of SGD-based methods to scenarios where dependence poses a challenge.

- **Robustness to Biased Gradients.** We show that biased SGD-based methods can converge and achieve accuracy comparable to unbiased ones, as long as the bias is not excessively large. This finding highlights the robustness of SGD-based methods in the presence of bias, broadening their applicability to scenarios where biased gradient estimates are encountered.

- **Accelerated Convergence with Polyak-Ruppert Averaging.** Our study reveals that incorporating Polyak-Ruppert averaging into SGD-based methods accelerates convergence (Polyak & Juditsky, 1992; Ruppert, 1988). Importantly, our findings emphasize the continued efficacy of this technique in challenging streaming settings characterized by dependence structures and biases. Furthermore, by combining Polyak-Ruppert averaging with the variance reduction capabilities offered by time-varying mini-batches, we obtain the best of both worlds. This powerful combination not only enhances the convergence rate but also increases robustness by reducing the variance.

Overall, our contributions extend the scope of SO by considering time-dependent and biased gradient estimates in a streaming framework. We provide non-asymptotic convergence rates, demonstrate the effectiveness of SGD-based methods in handling dependence and bias, and highlight the accelerated convergence achieved

through Polyak-Ruppert averaging. These findings enhance our understanding of SO in streaming settings and offer practical strategies for researchers and practitioners to optimize their models in real-time applications.

**Organization.** In section 2, we introduce the streaming framework and lay the groundwork for our non-asymptotic analysis. We provide an overview of key concepts, definitions, and assumptions, with a focus on the dependency structures. In section 2.1, we discuss the assumptions related to the objective function, while section 2.2 focuses on the assumptions concerning the gradient estimates, particularly the crucial role of $\alpha$-mixing conditions in validating the dependency structures. We then present our streaming variants of the SGD methods in section 2.3.

In section 3, we present our convergence results, considering both our streaming SGD variant with and without Polyak-Ruppert averaging. We provide a detailed discussion of each result, highlighting connections to previous work in the field. The proofs of our results are provided in appendix A.

Finally, in section 4, we present experimental results that validate our findings. We conduct experiments on synthetic and real-life time-dependent streaming data, showcasing the practical implications of our work.

## 2 Problem Formulation, Assumptions, and Methods

We consider SO problems of the form

$$\min_{\theta \in \mathbb{R}^d} \{F(\theta) = \mathbb{E}_\xi[f(\theta; \xi)]\}. \tag{1}$$

In the formulation (1), $\theta \in \mathbb{R}^d$ represents the parameter vector of interest. The objective function $F$ is defined as the expected value of the random loss function $f(\theta; \xi)$, which depends on the parameter vector $\theta$ and the random variable $\xi$. To minimize $F$, we rely on estimates of its gradients. Specifically, we estimate $F$ by evaluating the gradient estimates of $f$ on a sequence of samples $(\xi_t)$.

In our streaming setting, we have access to a sequence of time-varying mini-batches $(\xi_t)$, rather than knowing the true underlying distribution of $\xi$. Each $\xi_t$ consists of $n_t \in \mathbb{N}$ individual data points, represented by the mini-batch $\{\xi_{t,1}, \ldots, \xi_{t,n_t}\}$. We extend this notion of time-varying mini-batches by defining $f_t(\theta) = f(\theta; \xi_t)$. Consequently, $(f_t(\theta))$ constitutes a sequence of differentiable (possible non-convex) random loss functions (Nesterov et al., 2018; Nemirovskij & Yudin, 1983). Hence, each $f_t(\theta)$ comprises $n_t \in \mathbb{N}$ individual losses, represented by the mini-batch $\{f_{t,1}, \ldots, f_{t,n_t}\}$. For example, consider the scenario where $\xi_t$ represents a mini-batch of input-output pairs $\{(x_{t,i}, y_{t,i})\}_{i=1}^{n_t}$. In this case, for a model class $\{h_\theta\}_{\theta \in \Theta}$, we can express $f_{t,i}(\theta)$ as a combination of a loss function $l$ and a regularizer $\Omega$:

$$f_{t,i}(\theta) = l(y_{t,i}, h_\theta(x_{t,i})) + \Omega(\theta).$$

### 2.1 Assumptions on Objective Functions: Quasi-strong Convexity and Lipschitz Smoothness

In accordance with previous work by Bach & Moulines (2011); Gower et al. (2019); Nguyen et al. (2019), we adopt certain assumptions regarding the objective function $F$. Firstly, we assume that $F$ possesses a unique global minimizer $\theta^* \in \Theta$, where $\Theta$ is a closed convex set in $\mathbb{R}^d$. These assumptions align with techniques utilized in (online) convex optimization (Boyd & Vandenberghe, 2004; Nesterov et al., 2018; Hazan et al., 2016; Shalev-Shwartz et al., 2012). In addition, we assume that the objective function $F$ is $\mu$-quasi-strongly convex (Karimi et al., 2016; Necoara et al., 2019):

**Assumption 1** ($\mu$-quasi-strong convexity). *The objective function $F$ is differentiable with $\nabla_\theta F(\theta^*) = 0$ and there exists a constant $\mu > 0$ such that $\forall \theta \in \Theta$, we have*

$$F(\theta^*) \geq F(\theta) + \langle \nabla_\theta F(\theta), \theta^* - \theta \rangle + \frac{\mu}{2} \|\theta^* - \theta\|^2. \tag{2}$$

The $\mu$-quasi-strong convexity assumption serves as a relaxed version of strong convexity for the SO problem, providing a more conservative notion. Various objective functions $F$ employed in machine learning applications have been extensively investigated and documented by Teo et al. (2007). Researchers have also explored

milder degrees of convexity, such as the Polyak-Łojasiewicz condition (Polyak, 1963; Lojasiewicz, 1963) studied by Karimi et al. (2016); Gower et al. (2021) for SGD methods, and their Ruppert-Polyak average investigated by Gadat & Panloup (2023) under a Kurdyka-Łojasiewicz-type condition (Kurdyka, 1998; Lojasiewicz, 1963). Relaxing the assumption of strict convexity is essential in practice to ensure the robustness and adaptiveness of algorithms, particularly for non-strongly convex SO problems (Bach & Moulines, 2013; Nemirovski et al., 2009; Necoara et al., 2019; Khaled & Richtárik, 2023).

**$F$ being $\mu$-quasi-strongly convex does not guarantee the $\mu$-quasi-strong convexity of $(f_t)$.** It is crucial to understand that while the objective function $F$ satisfies the $\mu$-quasi-strong convexity assumption (2), the individual loss functions $(f_t)$ may not exhibit the same property. This distinction plays an essential role in our convergence analysis as it interacts with the time-dependency of the problem and the presence of biases. In section 3, we will explore this relationship in detail and investigate its implications for the convergence behavior. Specifically, we will examine how the level of dependence, biases, and the $\mu$-quasi-strong convexity conditions intertwine and affect the convergence properties of the optimization process.

To analyze the Polyak-Ruppert averaging estimate, we need to impose additional smoothness assumptions on the objective function $F$, following the framework established in Bach & Moulines (2011); Godichon-Baggioni et al. (2023). These smoothness assumptions ensure the necessary conditions for the convergence analysis of the Polyak-Ruppert method (Ruppert, 1988; Polyak & Juditsky, 1992).

**Assumption 2** ($C_\nabla$- and $C'_\nabla$-smoothness). *The objective function $F$ have $C_\nabla$-Lipschitz continuous gradients around $\theta^*$, i.e., there exists a constant $C_\nabla > 0$ such that $\forall \theta \in \Theta$,*

$$\|\nabla_\theta F(\theta) - \nabla_\theta F(\theta^*)\| \leq C_\nabla \|\theta - \theta^*\|. \tag{3}$$

*Next, the Hessian of $F$ is $C'_\nabla$-Lipschitz-continuous around $\theta^*$, that is, there exists a constant $C'_\nabla \geq 0$ such that $\forall \theta \in \Theta$,*

$$\|\nabla_\theta^2 F(\theta) - \nabla_\theta^2 F(\theta^*)\| \leq C'_\nabla \|\theta - \theta^*\|. \tag{4}$$

As highlighted in Bottou et al. (2018), the assumption stated in Assumption 2 guarantees that the gradient $\nabla_\theta F$ does not exhibit arbitrary variations. This property makes $\nabla_\theta F$ a valuable guide for reducing the value of $F$. In deterministic optimization, smooth optimization typically achieves faster convergence rates compared to non-smooth optimization. However, in the context of SO, the benefits of smoothness are limited to improvements in the associated constants (Nesterov et al., 2018).

## 2.2 Assumptions on Gradient Estimates: Dependence, Bias, Expected Smoothness, and Gradient Noise

Let $\mathcal{F}_t = \sigma(f_i : i \leq t)$ denote the natural filtration of the SO problem (1). The gradients of the time-varying mini-batches $(f_t(\theta))$ serve as estimates of $\nabla_\theta F(\theta)$. Unlike classical assumptions that typically demand unbiased (and uniformly bounded) gradient estimates (Godichon-Baggioni et al., 2023), we adopt a more flexible approach. We relax these constraints by allowing the gradients $(\nabla_\theta f_t(\theta))$ to be time-dependent and biased estimates of $\nabla_\theta F(\theta)$ in the following way:

**Assumption 3-p** ($D_\nu \nu_t$-dependence and $B_\nu \nu_t$-bias). *For each $t \geq 1$, the random function $\nabla_\theta f_t$ is square-integrable, $\mathcal{F}_t$-measurable, and for a positive integer $p$, there exists some positive sequence $(\nu_t)_{t \geq 1}$ and constants $D_\nu, B_\nu \geq 0$ such that*

$$\mathbb{E}[\|\mathbb{E}[\nabla_\theta f_t(\theta)|\mathcal{F}_{t-1}] - \nabla_\theta F(\theta)\|^p] \leq \nu_t^p (D_\nu^p \mathbb{E}[\|\theta - \theta^*\|^p] + B_\nu^p). \tag{5}$$

**Discussion of Assumption 3-p.** Assumption 3-p follows the form of mixing conditions for weakly dependent sequences, indicating that the dependence diminishes at a rate determined by $\nu_t$. The verification of Assumption 3-p can be accomplished using moment inequalities for partial sums of strongly mixing sequences (Rio, 2017), which is commonly referred to as *short-range dependence*; it is also known by other aliases such as *short memory* or *short-range persistence*.

Indeed, for any positive integer $p$, Assumption 3-p can be bounded above as follows:

$$\mathbb{E}[\|\mathbb{E}[\nabla_\theta f_t(\theta)|\mathcal{F}_{t-1}] - \nabla_\theta F(\theta)\|^p] \leq \mathbb{E}[\|\nabla_\theta f_t(\theta) - \nabla_\theta F(\theta)\|^p] = n_t^{-p} \mathbb{E}[\|S_t\|^p], \tag{6}$$

using Jensen's inequality. In (6), $S_t = \sum_{i=1}^{n_t} (\nabla_\theta f_{t,i}(\theta) - \nabla_\theta F(\theta))$ represents a $d$-dimensional vector, and $\nabla_\theta f_t(\theta) = n_t^{-1} \sum_{i=1}^{n_t} \nabla_\theta f_{t,i}(\theta)$. Let $(\nabla_\theta f_{t,i})$ be a strictly stationary sequence, and assume the existence of some $r > p$ such that $\sup_{x>0}(x^r Q(x))^{1/r} < \infty$, where $Q(x)$ denotes the quantile function of $\|\nabla_\theta f_{t,i}\|$. Suppose $(\nabla_\theta f_{t,i})$ is strongly $\alpha$-mixing according to Rosenblatt (1956), with strong mixing coefficients $(\alpha_t)_{t\geq 1}$ satisfying $\alpha_t = \mathcal{O}(t^{-pr/(2r-2p)})$. Then, by Rio (2017, Corollary 6.1), we find that $\mathbb{E}[\|S_t\|^p] = \mathcal{O}(n_t^{p/2})$. This implies that (6) is at most $\mathcal{O}(n_t^{-p/2})$. This encompasses several linear, non-linear, and Markovian time series, e.g., see Bradley (2005); Doukhan (2012) for more examples of other mixing coefficients of weak dependence and their relationships.

In relation to the form of Assumption 3-p, this indicates that $B_\nu \neq 0$ in this case. However, it is possible to have $B_\nu = 0$ in unbiased examples, as we will demonstrate later in section 4. Another example is the short-term dependent Markovian case, where $\nu_t = n_t^{-1/2}$, and the dependency constant $D_\nu$ in Assumption 3-p can be explicitly expressed using the mixing times of the Markov chain. Nagaraj et al. (2020) worked out precise calculations for least squares regression. To summarize, we encounter short-range dependence when (6) decays at most with $\mathcal{O}(n_t^{-p/2})$. Conversely, we encounter *long-range dependence* when (6) decays slower than $\mathcal{O}(n_t^{-p/2})$.

The classical convergence analysis for SGD-based methods relies on the assumption of (uniformly) bounded gradient estimates, which is restrictive and applicable only to certain types of losses (Bottou et al., 2018; Nguyen et al., 2018). In our work, we depart from this approach and instead adopt the assumptions introduced by Bach & Moulines (2011); Gower et al. (2019), who analyzed SGD for quasi-strongly convex objectives as expressed in (2). Specifically, we make assumptions regarding the expected smoothness of gradients $(\nabla_\theta f_t)$ and the expected finiteness of $(\nabla_\theta f_t(\theta^*))$:

**Assumption 4-p** ($C_\kappa$-expected smoothness). *For a positive integer $p$, there exists some positive constant $C_\kappa$ such that $\forall \theta \in \Theta$, $\mathbb{E}[\|\nabla_\theta f_t(\theta) - \nabla_\theta f_t(\theta^*)\|^p] \leq C_\kappa^p \mathbb{E}[\|\theta - \theta^*\|^p]$.*

**Assumption 5-p** ($\sigma_t$-gradient noise). *For a positive integer $p$, there exists some positive sequence $(\sigma_t)_{t\geq 1}$ such that $\mathbb{E}[\|\nabla_\theta f_t(\theta^*)\|^p] \leq \sigma_t^p$.*

As for Assumption 3-p, the verification of Assumption 5-p can be achieved using $\alpha$-mixing conditions and analogous arguments as those employed above, resulting in $\sigma_t^p$ being of the order $\mathcal{O}(n_t^{-p/2})$. It is worth noting that in the classical i.i.d. case with unbiased gradients, alternative relaxations of Assumptions 4-p and 5-p exist, such as the ER or ABC assumptions (Gower et al., 2021; Khaled & Richtárik, 2023). However, since our primary focus is on introducing time-dependent and biased gradient estimates, we do not delve into these alternative assumptions.

In summary, the assumptions we have presented (Assumptions 3-p to 5-p) offer a more relaxed framework compared to the standard assumptions found in the literature (Benveniste et al., 2012; Kushner & Yin, 2003; Bach & Moulines, 2011; Godichon-Baggioni et al., 2023; Dieuleveut et al., 2017; Dieuleveut & Bach, 2016). Our assumptions are designed to accommodate a wide range of scenarios, including both classical examples and more intricate models that involve learning from time-dependent data with biases. By adopting these assumptions, we provide a flexible and applicable framework for analyzing SO methods in such settings.

## 2.3 Stochastic Streaming Optimization Methods

To address the SO problem (1) within a streaming setting, we employ the Stochastic Streaming Gradient (SSG) method proposed by Godichon-Baggioni et al. (2023). The SSG algorithm is defined as follows

$$\textbf{(SSG)} \qquad \theta_t = \theta_{t-1} - \frac{\gamma_t}{n_t} \sum_{i=1}^{n_t} \nabla_\theta f_{t,i}(\theta_{t-1}). \qquad (7)$$

Here, $\gamma_t$ denotes the learning rate, which satisfies the conditions $\sum_{i=1}^{\infty} \gamma_i = \infty$ and $\sum_{i=1}^{\infty} \gamma_i^2 < \infty$ (Robbins & Monro, 1951). It is worth noting that when $n_t = 1$, the SSG algorithm reduces to the usual SGD method.

In many models, there are often constraints imposed on the parameter space, necessitating a projection of the parameters. To address this, we introduce the Projected Stochastic Streaming Gradient (PSSG) estimate,

---

**Algorithm 1:** Stochastic streaming gradient estimates (SSG/PSSG/ASSG/APSSG)

---

**Inputs** : $\theta_0 \in \Theta \subseteq \mathbb{R}^d$, project: **True** or **False**, average: **True** or **False**
**Outputs:** $\theta_t, \bar{\theta}_t$ (resulting estimates)
Initialization: $\bar{\theta}_0 \in \mathbb{R}^d$
**for** *each $t \geq 1$, a time-varying mini-batch of $n_t$ data arrives,* **do**

    $\theta_t \leftarrow \theta_{t-1} - \frac{\gamma_t}{n_t} \sum_{i=1}^{n_t} \nabla_\theta f_{t,i}(\theta_{t-1})$          /* update */

    **if** *project* **then**

        $\theta_t \leftarrow \mathcal{P}_\Theta(\theta_t)$          /* project */

    **if** *average* **then**

        $\bar{\theta}_t \leftarrow (N_{t-1}/N_t)\bar{\theta}_{t-1} + (n_t/N_t)\theta_{t-1}$          /* average */

---

defined as

$$\textbf{(PSSG)} \qquad \theta_t = \mathcal{P}_\Theta\left(\theta_{t-1} - \frac{\gamma_t}{n_t}\sum_{i=1}^{n_t} \nabla_\theta f_{t,i}(\theta_{t-1})\right), \qquad (8)$$

where $\mathcal{P}_\Theta$ represents the Euclidean projection onto $\Theta$, given by $\mathcal{P}_\Theta(\theta) = \arg\min_{\theta' \in \Theta}\|\theta - \theta'\|_2$.

If one were to examine the trajectory of the stochastic gradients $(\nabla_\theta f_{t,i})$, it would become apparent that they exhibit high noise levels and lack robustness, which can lead to slow convergence or even prevent convergence altogether. Therefore, it intuitively makes sense to use mini-batches of gradient estimates within each iteration, as this reduces variance and facilitates the adjustment of the learning rate $(\gamma_t)$, ultimately improving the quality of each iteration.

Within this streaming framework, we are interested in incorporating acceleration techniques into the existing algorithms presented in (7) and (8). One important extension is the Polyak-Ruppert procedure (Polyak & Juditsky, 1992; Ruppert, 1988), which ensures optimal statistical efficiency without compromising computational costs. The Averaged Stochastic Streaming Gradient (ASSG) method, denoted as (ASSG)/(APSSG), is defined by

$$\textbf{(ASSG)/(APSSG)} \qquad \bar{\theta}_t = \frac{1}{N_t}\sum_{i=0}^{t-1} n_{i+1}\theta_i. \qquad (9)$$

Here, $N_t = \sum_{i=1}^t n_i$ represents the accumulated sum of observations.[1] To ensure a fair comparison of our streaming methods, it is crucial to evaluate them in terms of the number of observations used, namely $N_t$. Similarly, we use APSSG to denote the (Polyak-Ruppert) averaged estimate of PSSG presented in (8). These averaging methods sequentially aggregate past estimates, resulting in smoother curves (i.e., variance reduction in the estimation trajectories), and accelerate convergence (Polyak & Juditsky, 1992).

The pseudo-code for these streaming estimates, including (7) to (9), is presented in algorithm 1. Additionally, (9) can be modified to a weighted average version, giving more weight to the latest estimates. This modification improves convergence while limiting the impact of poor initializations. Examples of such algorithms can be found in Boyer & Godichon-Baggioni (2022); Mokkadem & Pelletier (2011).

The popularity of SGD methods has prompted efforts to improve their efficiency, robustness, and user-friendliness. As a result, numerous variants, including second-order methods like Newton's method and other extensions, have been extensively explored and discussed in works such as Boyd & Vandenberghe (2004); Nesterov et al. (2018); Byrd et al. (2016).

The choice of the learning rate $(\gamma_t)$ significantly affects the convergence of SGD. A learning rate that is too small leads to slow convergence, while an excessively high learning rate may prevent convergence due to oscillations around the minimum of the loss function. Therefore, there is a strong motivation for the

---

[1]In practice, as we handle data sequentially, we employ the rewritten formula $\bar{\theta}_t = (N_{t-1}/N_t)\bar{\theta}_{t-1} + (n_t/N_t)\theta_{t-1}$, with $\bar{\theta}_0 = 0$. This allows us to update the averaged estimate efficiently.

development of adaptive learning rate mechanisms that require less manual fine-tuning and offer improved user-friendliness.[2] Moreover, the idea of having per-dimension learning rates that adjust individually as the convergence progresses holds potential advantages (Godichon-Baggioni & Tarrago, 2023).

In Bottou et al. (2018), a comprehensive overview of various SO methods is provided, covering both convex and non-convex optimization scenarios. The paper also delves into strategies for parallelization and distribution, aiming to accelerate the SGD updates and improve overall performance.

## 3 Convergence Analysis

In this section, we analyze the convergence of the streaming methods presented in (7) to (9). Our objective is to establish non-asymptotic bounds on $\delta_t = \mathbb{E}[\|\theta_t - \theta^*\|^2]$ and $\bar{\delta}_t = \mathbb{E}[\|\bar{\theta}_t - \theta^*\|^2]$, which solely depend on the SO problem parameters.

To achieve this, we derive a recursive relation for $\delta_t$ that allows us to non-asymptotically bound the estimates in (7) and (8) for any given sequences of $(\gamma_t)$, $(\nu_t)$, $(\sigma_t)$, and $(n_t)$:

**Lemma 1.** *Let $\delta_t = \mathbb{E}[\|\theta_t - \theta^*\|^2]$, where $(\theta_t)$ either follows the recursion in* (7) *or* (8). *Suppose Assumptions 1 and 3-p to 5-p hold for $p = 2$. Then, for any $(\gamma_t)$, $(\nu_t)$, $(\sigma_t)$, and $(n_t)$, we have*

$$\delta_t \leq [1 - (\mu - 2D_\nu \nu_t)\gamma_t + 2C_\kappa^2 \gamma_t^2]\delta_{t-1} + \frac{B_\nu^2}{\mu}\nu_t^2 \gamma_t + 2\sigma_t^2 \gamma_t^2. \tag{10}$$

To prove this lemma, we adapt classical techniques from stochastic approximations (Benveniste et al., 2012; Kushner & Yin, 2003), originally developed for the unbiased i.i.d. setting, to our specific streaming setting. Our adaptation incorporates the time-dependence and biases introduced by the new assumptions, Assumptions 3-p and 5-p. This enables us to gain novel insights into the interplay between dependence and biases, leading to a deeper understanding of the convergence properties of SO methods. Notably, the recursive relation for $\delta_t$ explicitly depends on $(\gamma_t)$, $(\nu_t)$, $(\sigma_t)$, and $(n_t)$, which, to the best of our knowledge, is a novel contribution. It highlights the connection between $\mu$-quasi-strong convexity and the dependence term $D_\nu v_t$, as discussed in section 2.1. We will later refer to this connection as *ensuring $\mu$-quasi-strong convexity through non-decreasing streaming batches* when the dependence sequence $(\nu_t)$ is linked to the time-varying mini-batches $(n_t)$.

In section 3.3, we will delve into the convergence analysis of the Polyak-Ruppert averaging estimate $\bar{\theta}_n$. Our goal is to establish a non-asymptotic bound on $\bar{\delta}_t$, which quantifies the convergence properties of the averaging method. This analysis builds upon the standard decomposition of the loss terms, enabling the emergence of the Cramér-Rao term (Bach & Moulines, 2011; Gadat & Panloup, 2023; Godichon-Baggioni et al., 2023). To facilitate this analysis, we also need to consider fourth-order moments. However, we will provide a comprehensive exploration of this aspect in detail in section 3.3.

### 3.1 Learning Rate $(\gamma_t)$, Uncertainty Terms $(\nu_t)$ and $(\sigma_t)$, and Time-Varying Mini-batches $(n_t)$

Before proceeding with the convergence analysis in sections 3.2 and 3.3, we first specify the functional forms of the learning rate $(\gamma_t)$, the uncertain terms $(\nu_t)$ and $(\sigma_t)$, and the streaming batch $(n_t)$.

**Learning Rate $(\gamma_t)$.** Following Godichon-Baggioni et al. (2023), we adopt the learning rate given by

$$\gamma_t = C_\gamma n_t^\beta t^{-\alpha},$$

where $C_\gamma > 0$, $\beta \in [0, 1]$, and $\alpha$ is chosen based on the expected streaming batches $n_t$. This learning rate allows us to assign more weight to larger streaming batches through the hyperparameter $\beta$.

**Uncertainty Terms $(\nu_t)$ and $(\sigma_t)$ from Assumptions 3-p and 5-p.** The uncertainty terms $(\nu_t)$ and $(\sigma_t)$ in our analysis play a crucial role in capturing the dependence and noise inherent in the SO problem. As

---

[2]E.g., see Momentum (Qian, 1999), Nesterov accelerated gradient (Nesterov, 1983), Adagrad (Duchi et al., 2011), Adadelta (Zeiler, 2012), RMSprop (Tieleman et al., 2012), and Adam (Kingma & Ba, 2014).

discussed in section 2.2, these terms can be viewed as functions of the streaming batches $(n_t)$. We define these terms as follows:

$$\nu_t = n_t^{-\nu} \quad \text{and} \quad \sigma_t = C_\sigma n_t^{-\sigma},$$

where $\nu \in (0, \infty)$, $\sigma \in [0, 1/2]$, and $C_\sigma > 0$. By setting $\nu_t = n_t^{-\nu}$, we introduce short-range dependence when $\nu \in [1/2, \infty)$ and long-range dependence when $\nu \in (0, 1/2)$. Hence, the classical i.i.d. case (Godichon-Baggioni et al., 2023) corresponds to $\nu \to \infty$. The choice of $\sigma \in [0, 1/2]$ aligns with the framework proposed by Godichon-Baggioni et al. (2023), where $\sigma = 1/2$ represents the i.i.d. case. When $\sigma < 1/2$, it allows for the presence of noisier outputs.

**Time-varying Mini-batches** $(n_t)$. Inspired by the work of Godichon-Baggioni et al. (2023), we adopt a formulation for time-varying mini-batches $(n_t)$ given by

$$n_t = \lceil C_\rho t^\rho \rceil,$$

where $C_\rho \in \mathbb{N}$ and $\rho \in [0, 1)$, ensuring that $n_t \in \mathbb{N}$. This formulation encompasses various scenarios, including classical (online) SGD methods for $C_\rho = 1$ and $\rho = 0$, (online) mini-batch SGD procedures with both constant and time-varying sizes when $C_\rho \in \mathbb{N}$ and $\rho = 0$ or $\rho \in [0, 1)$, respectively, as well as the Polyak-Ruppert average of (online) time-varying mini-batches. For ease of reference, we use the term *streaming batch* size to refer to $C_\rho$ and the term *streaming rate* for $\rho$.

### 3.2 Stochastic Streaming Gradients

**Theorem 1** (SSG/PSSG). *Let $\delta_t = \mathbb{E}[\|\theta_t - \theta^*\|^2]$, where $(\theta_t)$ either follows the recursion in (7) or (8). Suppose Assumptions 1 and 3-p to 5-p hold for $p = 2$. If $\mu_\nu = \mu - \mathbb{1}_{\{\rho=0\}} 2D_\nu C_\rho^{-\nu} > 0$, then there exist explicit constants $C_\delta, C'_\delta, C''_\delta > 0$ such that for $\alpha - \rho\beta \in (1/2, 1)$, we have*

$$\delta_t \leq \mathcal{O}\left( \exp\left( -\frac{\mu C_\gamma N_t^{\frac{1+\rho\beta-\alpha}{1+\rho}}}{C_\delta C_\rho^{\frac{1-\beta-\alpha}{1+\rho}}} \right) \right) + \frac{C'_\delta B_\nu^2}{\mu\mu_\nu C_\rho^{\frac{2\nu}{1+\rho}} N_t^{\frac{2\rho\nu}{1+\rho}}} + \frac{C''_\delta C_\sigma^2 C_\gamma}{\mu_\nu C_\rho^{\frac{2\sigma-\beta-\alpha}{1+\rho}} N_t^{\frac{\rho(2\sigma-\beta)+\alpha}{1+\rho}}}. \tag{11}$$

*An explicit version of this bound is given in appendix A.*

**Sketch of proof.** Under Assumptions 1 and 3-p to 5-p with $p = 2$, we can establish a recursive relation for $(\delta_t)$ defined by

$$\delta_t \leq [1 - (\mu - 2D_\nu\nu_t)\gamma_t + 2C_\kappa^2\gamma_t^2]\delta_{t-1} + \mu^{-1}B_\nu^2\nu_t^2\gamma_t + 2\sigma_t^2\gamma_t^2,$$

for any form of $(\gamma_t)$, $(\nu_t)$, $(\sigma_t)$, and $(n_t)$. The non-asymptotic upper bound of this recursive relation (10) can be explicitly derived using classical techniques from stochastic approximations (Benveniste et al., 2012; Kushner & Yin, 2003). Bounding the projected estimate in (8) can directly be obtained by noting that $\mathbb{E}[\|\mathcal{P}_\Theta(\theta) - \theta^*\|^2] \leq \mathbb{E}[\|\theta - \theta^*\|^2]$. Alternatively, the projected estimate (8) can also be proved by assuming a bounded gradient, which replaces Assumptions 4-p and 5-p. Detailed discussions and alternative proofs can be found in works such as Bach & Moulines (2011); Godichon-Baggioni et al. (2023). The discontinuity in $\rho$ emerges from the introduction of an indicator that determines whether $(\nu_t)$ is constant or decreasing. When $(\nu_t)$ is constant, it becomes challenging to obtain a bound that closely approaches $\mu$. Conversely, when $(\nu_t)$ decreases, there is a possibility of achieving a bound that can come as close as possible to $\mu$. In the i.i.d. case, this choice corresponds to the optimal strong convexity parameter (Godichon-Baggioni et al., 2023). Alternatively, we can derive a better bound by considering a continuous trade-off where we minimize $(\nu_t)$ in the bound while minimizing the other terms. This continuous trade-off has the potential to provide a smoother transition between the cases of constant and decreasing $(\nu_t)$.

**Related work.** Our work extends and aligns with previous studies in the field. First, our results replicate the outcomes of the unbiased i.i.d. case investigated in Godichon-Baggioni et al. (2023), specifically when $B_\nu = 0$ and $\sigma = 1/2$. This demonstrates the consistency of our findings with regard to the unbiased i.i.d. setting. Additionally, our results are in line with the research conducted by Bach & Moulines (2011), which focused on the unbiased i.i.d. case in a non-streaming setting, i.e., when $C_\rho = 1$ and $\rho = 0$.

**Bound on objective function.** If the objective function $F$ has gradients that are $C_\nabla$-Lipschitz continuous (Assumption 2), the inequality given by (11) provides a bound on the function values of $F$. Specifically, we have $\mathbb{E}[F(\theta_t) - F(\theta^*)] \leq C_\nabla \delta_t/2$ by applying Cauchy-Schwarz's inequality.

**Ensuring $\mu$-quasi-strong convexity through non-decreasing streaming batches.** The positivity of the dependence penalized convexity constant $\mu_\nu = \mu - \mathbb{1}_{\{\rho=0\}} 2D_\nu C_\rho^{-\nu}$ is crucial for all terms of (11). This constraint arises from the fact that while the objective function $F$ is $\mu$-quasi-strongly convex, the loss functions $(f_t)$ may not possess the same convexity properties, e.g., see discussion in section 2.1. This is demonstrated in sections 4.1.2 and 4.1.3 for ARCH models (Werge & Wintenberger, 2022).

So, how should we interpret $\mu_\nu$? When the streaming rate $\rho = 0$, the positivity of $\mu_\nu$ depends on the convexity constant $\mu$, the streaming batch size $C_\rho$, and the imposed dependencies specified in Assumption 3-p. If the dependence quantity $D_\nu$ is sufficiently large that $\mu_\nu$ becomes non-positive, it becomes necessary to choose a larger streaming batch size $C_\rho$ to ensure $\mu_\nu$ remains positive. In other words, strong dependency structures reduce convexity, but large mini-batches counteracts this effect.

An alternative approach to ensuring convexity is by employing increasing streaming batches, i.e., setting the streaming rate $\rho > 0$. This method provides greater robustness as we no longer need to determine a specific value for the mini-batch size $C_\rho$ to maintain the positivity of $\mu_\nu$. However, combining both approaches is ideal as it ensures convexity while a larger $C_\rho$ effectively reduces variance, resulting in a more favorable outcome.

In section 4, we delve into the challenge of calibrating $C_\rho$ and $\rho$ appropriately to achieve both stability and convergence in the optimization process. The goal of this calibration is to strike a balance between taking frequent gradient steps for faster convergence and ensuring the positivity of $\mu_\nu$. To accomplish this, it is crucial to choose a sufficiently large $C_\rho$ and $\rho$ that can maintain the positivity of $\mu_\nu$ while allowing for as many gradient steps as possible. This delicate balance is essential in achieving optimal performance in the optimization process.

**Variance reduction with larger streaming batches $C_\rho$.** Unsurprisingly, larger streaming batches $C_\rho$ have a variance-reducing effect, as illustrated in section 4. This effect is explicitly demonstrated in each term of (11). Interestingly, the variance reduction resulting from large mini-batches scales with batch size but does not increase the decay rate of $\delta_t$.

**Decay of initial conditions.** The initial conditions in the first term of (11) decay sub-exponentially. A detailed expression for this term can be found in appendix A. The last term of (11) represents the noise term, which is influenced by the gradient noise as described in Assumption 5-p. When $\alpha - \rho\beta \in (1/2, 1)$, the noise term decays at a rate of $\mathcal{O}(N_t^{-(\rho(2\sigma-\beta)+\alpha)/(1+\rho)})$. For instance, if we set $\alpha = 2/3$, $\beta = 1/3$, and $\sigma = 1/2$, the noise term decays as $\mathcal{O}(N_t^{-2/3})$ for any streaming rate $\rho \in [0, 1)$. This decay behavior is illustrated in Godichon-Baggioni et al. (2023). Notably, in the unbiased cases ($B_\nu = 0$), this noise term corresponds to the asymptotic term (Godichon-Baggioni et al., 2023). Moreover, when $\alpha + \beta < 2\sigma$, the noise/asymptotic term is positively influenced by larger streaming batches $C_\rho$. This effect is demonstrated in section 4.

**Behavior under biasedness, $B_\nu > 0$.** The second term in (11) represents a pure bias term determined by the bias quantity $B_\nu$, the level of dependence $\nu$, and the (dependence-penalized) convexity constant $\mu_\nu$. Importantly, the bias term is independent of the learning rate $\gamma_t = C_\gamma n_t^\beta t^{-\alpha}$, but depends on the time-varying mini-batches $n_t = C_\rho t^\rho$, i.e., through the streaming batch size $C_\rho$ and the streaming rate $\rho$.

The dependence term exhibits a scaling of $\mathcal{O}(N_t^{-2\rho\nu/(1+\rho)})$. For instance, to achieve a decay rate of $\mathcal{O}(N_t^{-1/2})$, we would need $\rho = 1$ and $\nu = 1/2$. It is remarkable that Theorem 1 accommodates both long- and short-range dependence. While long-range dependence leads to slow convergence (slower than $\mathcal{O}(N_t^{-1/2})$), a positive streaming rate $\rho$ can break long-range dependence. In summary, by increasing the streaming batch size, we preserve $\mu$-quasi-strong convexity and alleviate both long- and short-term dependence. This results in a bound of $\delta_t = \mathcal{O}(\max\{\mathbb{1}_{\{B_\nu > 0\}} N_t^{-2\rho\nu/(1+\rho)}, N_t^{-(\rho(2\sigma-\beta)+\alpha)/(1+\rho)}\})$.

### 3.3 Averaged Stochastic Streaming Gradients

In our subsequent analysis, our main focus is on the averaging estimate $\bar{\theta}_n$, which is defined in (9). This averaging estimate is obtained from either the SSG estimate in (7) or the PSSG estimate in (8). Our analysis relies on the standard decomposition of the loss terms, which allows the Cramér-Rao term to emerge (Bach & Moulines, 2011; Gadat & Panloup, 2023; Godichon-Baggioni et al., 2023). To facilitate this analysis, it is necessary to consider the fourth-order moments, which require the assumptions Assumptions 3-p to 5-p to hold for $p = 4$. Moreover, based on Assumption 5-p, where $\sigma_t = C_\sigma n_t^{-\sigma}$ with $\sigma \in [0, 1/2]$, we introduce an additional assumption regarding the covariance of the score vectors associated with the parameter vector $\theta^*$.

**Assumption 6** (Covariance of the scores). *There exists a non-negative self-adjoint operator $\Sigma$ such that $\forall t \geq 1$, $n_t^{2\sigma}\mathbb{E}[\nabla_\theta f_t(\theta^*)\nabla_\theta f_t(\theta^*)^\top] \preceq \Sigma + \Sigma_t$, where $\Sigma_t$ is a positive symmetric matrix with $Tr(\Sigma_t) = C'_\sigma n_t^{-2\sigma'}$, $C'_\sigma \geq 0$, and $\sigma' \in (0, 1/2]$.*

In some cases, such as the i.i.d. scenario and certain unbiased situations discussed in section 4.1.1, Assumption 6 holds with $\sigma = 1/2$ and $C'_\sigma = 0$ (Godichon-Baggioni et al., 2023). This condition applies to scenarios characterized by short-range dependence, as shown in section 4.1.1 with $\sigma = 1/2$ (and $C'_\sigma > 0$). Conversely, the long-range dependence case occurs when $\sigma < 1/2$ (and $C'_\sigma > 0$). Under Assumption 6, we can derive the dominant term $\Lambda/N_t$, where $\Lambda = \text{Tr}(\nabla_\theta^2 F(\theta)^{-1}\Sigma\nabla_\theta^2 F(\theta)^{-1})$. This assumption also enables the establishment of the Cramer-Rao lower bound for the case of unbiased i.i.d. samples. Specifically, the bound can be expressed as

$$\mathbb{E}[\|\bar{\theta}_t - \theta^*\|^2] \leq \mathcal{O}(\Lambda N_t^{-1}) + \mathcal{O}(N_t^{-b}) \quad \text{with} \quad b > 1.$$

In order to analyze the projected averaged estimate (a.k.a. APSSG), an additional assumption is made to avoid the computation of the sixth-order moment. We assume that $(\nabla_\theta f_t)$ is uniformly bounded on the compact $\Theta$. However, it is worth mentioning that the derivation of the sixth-order moment can be found in Godichon-Baggioni (2016).

**Assumption 7** ($G_\Theta$-bounded gradients). *Let $D_\Theta = \inf_{\theta \in \partial\Theta}\|\theta - \theta^*\| > 0$ with $\partial\Theta$ denoting the boundary of $\Theta$. Moreover, there exists $G_\Theta > 0$ such that $\forall t \geq 1$, $\sup_{\theta \in \Theta}\|\nabla_\theta f_t(\theta)\|^2 \leq G_\Theta^2$ a.s.*

**Theorem 2** (ASSG/PASSG). *Let $\bar{\delta}_t = \mathbb{E}[\|\bar{\theta}_t - \theta^*\|^2]$ with $\bar{\theta}_n$ given by (9), where $(\theta_t)$ either follows the recursion in (7) or (8). Suppose Assumptions 1 to 6 hold for $p = 4$. In addition, Assumption 7 must hold if $(\theta_t)$ follows the recursion in (8). If $\mu_\nu = \mu - \mathbb{1}_{\{\rho=0\}}2D_\nu C_\rho^{-\nu} > 0$, then for $\alpha - \rho\beta \in (1/2, 1)$, we have*

$$\bar{\delta}_t^{\frac{1}{2}} \leq \frac{\Lambda^{\frac{1}{2}}}{N_t^{\frac{1}{2}}}\mathbb{1}_{\{\sigma=1/2\}} + \frac{2^{\frac{1}{2}}\Lambda^{\frac{1}{2}}C_\rho^{\frac{1-2\sigma}{2(1+\rho)}}}{N_t^{\frac{1+2\rho\sigma}{2(1+\rho)}}}\mathbb{1}_{\{\sigma<1/2\}} + \frac{2^{\frac{1}{2}}C_\sigma'^{\frac{1}{2}}C_\rho^{\frac{1-2(\sigma+\sigma')}{2(1+\rho)}}}{\mu N_t^{\frac{1+2\rho(\sigma+\sigma')}{2(1+\rho)}}} \tag{12}$$

$$+\tilde{\mathcal{O}}\left(\max\left\{N_t^{-\frac{2+\rho(2\sigma+\beta)-\alpha}{2(1+\rho)}}, N_t^{-\frac{\rho(2\sigma-\beta)+\alpha}{1+\rho}}\right\}\right) + \mathbb{1}_{\{B_\nu \neq 0\}}\Psi_t, \tag{13}$$

*with*

$$\Psi_t = \tilde{\mathcal{O}}\left(\max\left\{N_t^{-\frac{\rho(\sigma+\nu)}{2(1+\rho)}}, N_t^{-\frac{1+\rho(\beta+\nu)-\alpha}{1+\rho}}, N_t^{-\frac{1+2\rho\nu}{2(1+\rho)}}, N_t^{-\frac{\delta/2+\rho\nu}{2(1+\rho)}}, N_t^{-\frac{2\rho\nu}{1+\rho}}\right\}\right),$$

*where $\delta = \mathbb{1}_{\{B_\nu=0\}}(\rho(2\sigma - \beta) + \alpha) + \mathbb{1}_{\{B_\nu \neq 0\}}\min\{\rho(2\sigma - \beta) + \alpha, 2\rho\nu\}$. An explicit version of this bound is given in appendix A.*

**Related work.** It is important to note that the bound presented in Theorem 2 is for the root mean square error, while our discussions focus on $\bar{\delta}_t$ without taking the root. Similar to the unbiased i.i.d. case explored in Godichon-Baggioni et al. (2023), the dominant term of $\bar{\delta}_t$ in (12) and (13) is $\Lambda/N_t$, which achieves the asymptotically optimal Cramer-Rao lower bound (Murata & Amari, 1999). Each term in (12) directly stems from Assumption 6. Importantly, these terms remain unaffected by the choice of learning rate ($\gamma_t$), but depend on the time-varying mini-batches ($n_t$). As discussed in Gadat & Panloup (2023), the bound of $\bar{\delta}_t$ can be interpreted as a bias-variance decomposition between the leading terms in (12) and the remaining terms in (13).

**Ensuring $\mu$-quasi-strong convexity through non-decreasing streaming batches.** The interpretation of $\mu_\nu$ in Theorem 2 is similar to that in Theorem 1. In both cases, the positivity of $\mu_\nu$ plays a crucial role in

all terms of (13), despite not being immediately apparent. Analogous to the insights gained from Theorem 1, increasing the streaming batch size allows us to preserve $\mu$-quasi-strong convexity and alleviate both long- and short-term dependence. By choosing a larger streaming batch size, we ensure the positivity of $\mu_\nu$, as demonstrated in the experiments conducted for ARCH models in sections 4.1.2 and 4.1.3.

**Accelerated decay through Polyak-Ruppert averaging.** Through Polyak-Ruppert averaging, it is possible to achieve the leading term $\Lambda/N_t$, which is known to attain the asymptotically optimal Cramer-Rao bound in the i.i.d. case (Godichon-Baggioni et al., 2023). Consequently, we can achieve the optimal and irreducible rate of $\bar{\delta}_t = \mathcal{O}(N_t^{-1})$. This is always accomplished in the unbiased case ($B_\nu = 0$) with $\sigma = 1/2$, even under short-range dependence.

In the specific case of $\sigma = 1/2$, (12) and (13) simplify significantly. The last two terms of (12) become negligible as $\sigma' > 0$. The first term of (13) decays at the rate $\mathcal{O}(N_t^{-(2+\rho(2\sigma+\beta)-\alpha)/(1+\rho)})$ or $\mathcal{O}(N_t^{-2(\rho(2\sigma-\beta)+\alpha)/(1+\rho)})$. Choosing $\alpha, \beta$ such that $\alpha + \rho(2\sigma/3 - \beta) = 2/3$ (e.g., $\alpha = 2/3$, $\beta = 1/3$, and $\sigma = 1/2$) yields a decay of $\mathcal{O}(N_t^{-4/3})$ for any $\rho$. Hence, by setting $\alpha = 2/3$ and $\beta = 1/3$ when $\sigma = 1/2$, the first term of (13) robustly achieves $\mathcal{O}(N_t^{-4/3})$ for any streaming rate $\rho$. Similarly, the last term of (13) is $\mathcal{O}(N_t^{-\rho(1/2+\nu)/(1+\rho)})$ for any $\rho$ when $\alpha = 2/3$ and $\beta = 1/3$. In summary, Theorem 2 with $\alpha = 2/3$, $\beta = 1/3$, and $\sigma = 1/2$ simplifies to the bound:

$$\bar{\delta}_t^{\frac{1}{2}} \leq \frac{\Lambda^{\frac{1}{2}}}{N_t^{\frac{1}{2}}} + \tilde{\mathcal{O}}\left(N_t^{-\frac{2}{3}}\right) + \mathbb{1}_{\{B_\nu \neq 0\}}\tilde{\mathcal{O}}\left(N_t^{-\frac{\rho(1/2+\nu)}{2(1+\rho)}}\right). \tag{14}$$

**Variance reduction from larger streaming batches** $C_\rho$**.** Polyak-Ruppert averaging accelerates convergence and, in particular, the decay rate. However, by taking large mini-batches, we can also achieve variance reduction. As discussed earlier (for Theorem 1), larger mini-batches scale the error, which is particularly beneficial in the initial stages. Therefore, combining averaging and mini-batches offers the best of both worlds, resulting in a better slope (decay rate) and intercept.

**Behavior under biasedness,** $B_\nu > 0$**.** The bias term $B_\nu$ affects only $\Psi_t$, except for the second term in (13), which impacts the decay rate $\delta$. Increasing the streaming rate $\rho$ diminishes the negative influence of the bias term. Surprisingly, $\Psi_t$ approaches zero as $t$ increases indefinitely for any $\nu$. However, achieving the desired decay rate of $\bar{\delta}_t = \mathcal{O}(N_t^{-1})$ excludes long-range dependence. Nevertheless, in our experiments (see section 4), the bias term does not seem to have a significant impact.

## 4 Experiments

In this section, we present the details and results of our experiments, aimed at evaluating the performance of our streaming SGD-based methods on both synthetic and real-world data. Our objective is to demonstrate findings that highlight the following: (i) time-varying mini-batch SGD methods have the capability to break long- and short-range dependence structures, (ii) biased SGD methods can achieve comparable performance to their unbiased counterparts, and (iii) incorporating Polyak-Ruppert averaging can accelerate the convergence. These findings collectively showcase the effectiveness and potential of our streaming SGD-based methods.

### 4.1 Synthetic Data

A way to illustrate our findings is by use of classical methods that aim to model and predict an underlying sequence of real-valued time-series $(X_s)$; here $s$ is short notation for indexing the sequence of observations, $(X_{N_t}, X_{N_t-1}, \ldots, X_{N_t-n_t} \equiv X_{N_{t-1}}, X_{N_{t-1}-1}, \ldots)$ with $N_t = \sum_{i=1}^t n_t$. The AutoRegressive (AR), Moving-Average (MA), and AutoRegressive Moving-Average (ARMA) models are the most well-known models for time-series (Brockwell & Davis, 2009; Box et al., 2015; Hamilton, 2020). The standard time-series analysis often relies on independence and constant noise, but it can be relaxed by, e.g., the AutoRegressive Conditional Heteroskedasticity (ARCH) model (Engle, 1982). Online learning algorithms of (both stationary and non-stationary) dependent time-series have been studied in Agarwal & Duchi (2012); Anava et al. (2013); Wintenberger (2021).

### 4.1.1 AR Model

A process $(X_s)$ is called a (zero-mean) AR(1) process, if there exists real-valued parameter $\theta$ such that $X_s = \theta X_{s-1} + \epsilon_s$, where $(\epsilon_s)$ is weak white noise with zero mean and variance $\sigma_\epsilon^2$. To illustrate the versatility of our results, we construct some (heavy-tailed) noise processes with long-range dependence: the noisiness is integrated using a Student's $t$-distribution with degrees of freedom above four, denoted by $(z_s)$. The long-range dependence is incorporated by multiplying $(z_s)$ with the fractional Gaussian noise $G_s(H) = B_{s+1}(H) - B_s(H)$, where $(B_s(H))$ is a fractional Brownian motion with Hurst index $H \in (0, 1)$. $(G_s(H))$ can also be seen as a (zero-mean) Gaussian process with stationary and self-similar increments (Nualart, 2006).

**Well-specified case.** Consider the well-specified case, in which, we estimate an AR(1) model from the underlying stationary AR(1) process $X_s = \theta^* X_{s-1} + \epsilon_s$ with $|\theta^*| < 1$. The squared loss function $f_t(\theta) = n_t^{-1} \sum_{i=1}^{n_t} (X_{N_{t-1}+i} - \theta X_{N_{t-1}+i-1})^2$ with gradient $\nabla_\theta f_t(\theta) = -2n_t^{-1} \sum_{i=1}^{n_t} X_{N_{t-1}+i-1}(X_{N_{t-1}+i} - \theta X_{N_{t-1}+i-1})$. Thus, the objective function is $F(\theta) = (\sigma_\epsilon^2(\theta^* - \theta)^2)/(1 - (\theta^*)^2) + \sigma_\epsilon^2$, as $\mathbb{E}[X_s] = 0$ and $\mathbb{E}[X_s^2] = \sigma_\epsilon^2/(1 - (\theta^*)^2)$, yielding $\nabla_\theta F(\theta) = 2\sigma_\epsilon^2(\theta - \theta^*)/(1 - (\theta^*)^2)$. Assumption 3-p with $p = 2$ can be written as follows:

$$\mathbb{E}[\|\mathbb{E}[\nabla_\theta f_t(\theta)|\mathcal{F}_{t-1}] - \nabla_\theta F(\theta)\|^2] = \frac{4(\theta - \theta^*)^2(1 - (\theta^*)^{2n_t})^2 \sigma_\epsilon^2}{(1 - (\theta^*)^2)^4 n_t^2} \left( \sigma_\epsilon^2 + \frac{1}{1 - (\theta^*)^2} \right).$$

This implies that Assumption 3-p holds when $(X_s)$ has bounded moments, which is satisfied under the natural constraint $|\theta^*| < 1$.[3] As a result, we can deduce that $D_\nu > 0$, $B_\nu = 0$, and $\nu_t$ is $\mathcal{O}(n_t^{-1})$. Similarly, Assumption 5-p can be verified in the same manner, with $\sigma_t$ being $\mathcal{O}(n_t^{-1/2})$. Additionally, Assumption 2 holds with $C_\nabla = 2\sigma_\epsilon^2/(1 - (\theta^*)^2)$ and $C'_\nabla = 0$, while Assumption 6 holds with $\Sigma = 4\sigma_\epsilon^4/(1 - (\theta^*)^2)$ and $\Sigma_t = 0$. Furthermore, for an AR(1) process $(X_s)$ constructed using the noise process $\epsilon_s = \sqrt{G_s(H)} z_s$ with Hurst index $H \geq 1/2$, one can verify that $\nu_t^4$ and $\sigma_t^4$ are $\mathcal{O}(n_t^{H-1})$ using the self-similarity property (Nourdin, 2012).

**Misspecified case.** Now, assume that the underlying data generating process follows a MA(1)-process: $X_s = \epsilon_s + \phi^* \epsilon_{s-1}$, with $\phi^* \in \mathbb{R}$. The misspecification error of fitting an AR(1) model to a MA(1) process can be found by minimizing $F(\theta) = \mathbb{E}[(X_s - \theta X_{s-1})^2] = \sigma_\epsilon^2(1 + (\phi^* - \theta)^2 + \theta^2(\phi^*)^2)$, where $\nabla_\theta F(\theta) = 2(\theta - \phi^*)\sigma_\epsilon^2 + 2\theta(\phi^*)^2\sigma_\epsilon^2$. Thus, as $\theta^* = \arg\min_\theta F(\theta) \equiv \arg\min_\theta (\phi^* - \theta)^2 + \theta^2(\phi^*)^2$ is a strictly convex function in $\theta$, we have $\nabla_\theta F(\theta) = 0 \Leftrightarrow 2(\theta - \phi^*) + 2\theta(\phi^*)^2 = 0 \Leftrightarrow 2\theta(1 + (\phi^*)^2) = 2\phi^* \Leftrightarrow \theta = \phi^*/(1 + (\phi^*)^2)$. Thus, for any $\phi^* \in \mathbb{R}$, we have $\theta \in (-1/2, 1/2)$. With this in mind, we can conduct our study of fitting an AR(1) model to a MA(1) process with $\phi^*$ drawn randomly from $\mathbb{R}$ (figure 1b). Furthermore, this reparametrization trick can be used to verify Assumption 3-p in the following way:

$$\mathbb{E}[\|\mathbb{E}[\nabla_\theta f_t(\theta)|\mathcal{F}_{t-1}] - \nabla_\theta F(\theta)\|^2] = \frac{4(\theta - \theta^*)^2}{n_t^2} f_{\phi^*}(\epsilon_{N_{t-1}}),$$

where $f_{\phi^*}(\epsilon_{N_{t-1}})$ is finite function depending on the moments of $(\epsilon_{N_{t-1}})$ and $\phi^*$.[4] Consequently, we establish $D_\nu > 0$, $B_\nu = 0$, and $\nu_t$ being $\mathcal{O}(n_t^{-1})$. Similarly, using the reparametrization trick, we can verify that $\sigma_t$ is $\mathcal{O}(n_t^{-1/2})$ (Assumption 5-p).

### 4.1.2 ARCH Model

In time series analysis, modeling heteroscedasticity of the conditional variance is a crucial component, especially in capturing phenomena like volatility clustering in financial time series. ARCH models are widely recognized models that effectively incorporate this characteristic. A process $(\epsilon_s)$ is called an ARCH(1) process with parameters $\alpha_0$ and $\alpha_1$ if it satisfies

$$\begin{cases} \epsilon_s = \sigma_s z_s, \\ \sigma_s^2 = \alpha_0 + \alpha_1 \epsilon_{s-1}^2, \end{cases} \tag{15}$$

where $\alpha_0 > 0$ and $\alpha_1 \geq 0$ ensures the non-negativity of the conditional variance process $(\sigma_s^2)$, and the innovations $(z_s)$ is white noise. We employ the Quasi-Maximum Likelihood (QML) procedure for the

---

[3]The verification is available in a longer version in appendix B.
[4]The verification is available in a longer version in appendix B.

statistical inference as outlined in Werge & Wintenberger (2022); the quasi likelihood losses is given by $f_s(\theta) = 2^{-1}(\epsilon_s^2/\sigma_s^2(\theta) + \log(\sigma_s^2(\theta)))$ with first-order derivative

$$\nabla_\theta f_s(\theta) = \nabla_\theta \sigma_s^2(\theta) \left( \frac{\sigma_s^2(\theta) - \epsilon_s^2}{2\sigma_s^4(\theta)} \right),$$

where $\nabla_\theta \sigma_s^2(\theta) = (1, \epsilon_{s-1}^2)^T$. Verification of Assumptions 3-p to 5-p can be done using mixing conditions; Francq & Zakoian (2019, Theorem 3.5) showed that stationary ARCH processes are geometrically $\beta$-mixing, which implies $\alpha$-mixing as well. Observe that the loss function ($f_s$) itself is not strongly convex but only the objective function $F$ may be strongly convex; convexity conditions of ARCH processes was investigated in Wintenberger (2021). This makes the parameters challenging to estimate in empirical applications as the optimization algorithms can quickly fail or converge to irregular solutions. There are different ways to overcome lack of convexity: first, projecting the estimates such that the (conditional) variance process ($\sigma_s^2$) stays away from zero (and close to the unconditional variance). Second, in the specific example of ARCH model, one could also recover convexity by implementing variance targeting techniques; an example using Generalized ARCH (GARCH) models can be found in Werge & Wintenberger (2022). To simplify our analysis we consider stationary ARCH(1) processes, where we fix $\alpha_0$ at 1 and initialize it at $1/2$.

### 4.1.3 AR-ARCH Model

We complete our experiments by considering an AR models with ARCH noise: the process ($X_s$) is called an AR(1)-ARCH(1) process with parameters $\theta$, $\alpha_0$, and $\alpha_1$, if it satisfies

$$\begin{cases} X_s = \theta X_{s-1} + \epsilon_s, \\ \epsilon_s = \sigma_s z_s, \\ \sigma_s^2 = \alpha_0 + \alpha_1 \epsilon_{s-1}^2, \end{cases} \tag{16}$$

where the innovations ($z_s$) is weak white noise. The statistical inference of this model is done using the squared loss for the AR-part and the QML procedure for the ARCH part, e.g., see sections 4.1.1 and 4.1.2. Assumptions 3-p and 5-p can be verified by Doukhan (1994, Proposition 6), which showed that ARMA-ARCH processes are $\beta$-mixing.

### 4.1.4 Discussion

Our experimental evaluation assesses the performance using the mean quadratic error $\mathbb{E}[\|\theta_{N_t} - \theta^*\|^2]$ across one thousand replications. The initial values $\theta_0$ and $\theta$ are randomly generated according to the specifications of the models. We intentionally omit projecting our estimates to highlight the errors stemming from dependence and lack of convexity. By averaging over multiple replications, we observe a reduction in variability, which primarily benefits the SSG method. The experiments aim to demonstrate the impact of the choice of $C_\rho$ and $\rho$ on the measures of dependence $D_\nu$, bias $B_\nu$, and the penalized convexity constant $\mu_\nu$ associated with dependence. To facilitate a comparison between different data streams, we set the parameters $C_\gamma = 1$, $\alpha = 2/3$, and $\beta = 0$ as fixed values.

The experiments described in sections 4.1.1 to 4.1.3 can be found in figure 1; here $\{C_\rho = 1, \rho = 0\}$ corresponds to the classical SG descent and its (Polyak-Ruppert) average estimate, $\{C_\rho = 64, \rho = 0\}$ is a mini-batch SSG/ASSG, and $\{C_\rho = 64, \rho = 1/2\}$ is an increasing SSG/ASSG with initial batch size of $C_\rho = 64$.

Let's examine the AR cases, both well-specified and misspecified, as shown in figures 1a and 1b. These figures display the results for long-range dependent white noise processes with a Hurst index of $H = 3/4$. It is evident that each pair of data streams converges, but the traditional SG method exhibits significant initial noise, particularly affecting the average estimate period without affecting its decay rate. An improvement could be achieved by modifying our average estimate to a weighted average version, which could mitigate the impact of poor initializations (Mokkadem & Pelletier, 2011; Boyer & Godichon-Baggioni, 2022). Nonetheless, even with this noise, the ASSG method still demonstrates better convergence. Both methods show a noticeable reduction in variance as $C_\rho$ increases, which is particularly advantageous in the early stages. However, excessively large streaming batch sizes $C_\rho$ may hinder convergence due to fewer iterations. Additionally,

Figure 1: Simulation of various data streams $n_t = C_\rho t^\rho$. See section 4.1 for details.

(a) AR(1): well-specified case. See section 4.1.1 for details. (b) AR(1): misspecified case. See section 4.1.1 for details.

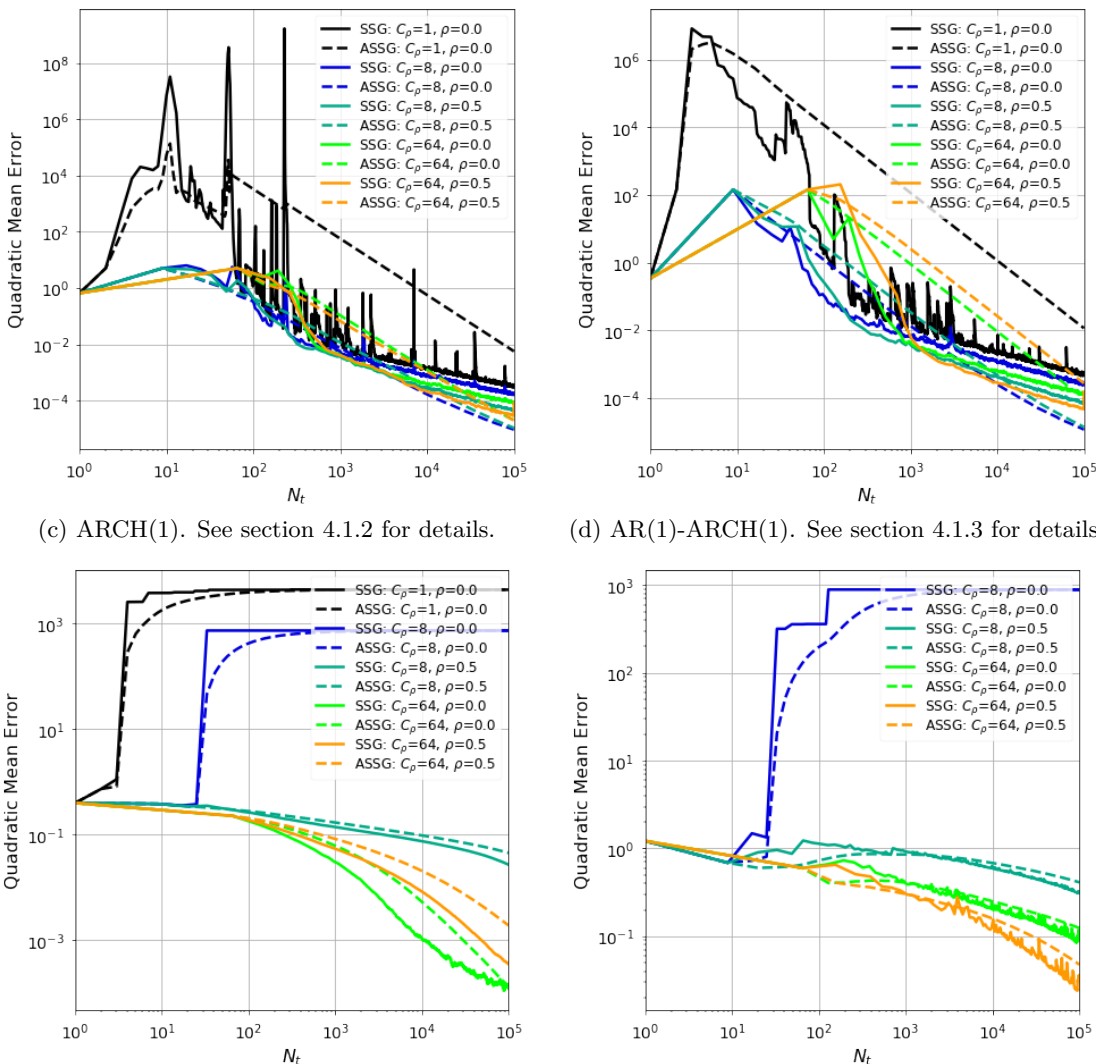

(c) ARCH(1). See section 4.1.2 for details. (d) AR(1)-ARCH(1). See section 4.1.3 for details.

figures 1a and 1b indicate improved decay for the SSG methods as the streaming rate $\rho$ increases. On the other hand, improvements in the ASSG method are not observed, as we do not leverage the potential of using more observations through the $\beta$ parameter, which could accelerate convergence (refer to section 4.2). It is surprising that we do not observe any effects from the long-range dependent white noise processes, but this seems to be an artifact effect in the proof, as fourth-order moments are required (i.e., Assumptions 3-p to 5-p with $p = 4$). Therefore, we conjecture that only second-order moment properties are responsible for the behavior of our simulations and that $\sigma = 1/2$ holds even for long-range dependent white noise processes, as proven in section 4.1.1.

Moving on to figures 1c and 1d, we present the experiments for stationary ARCH(1) models, both with and without an AR component, as outlined in sections 4.1.2 and 4.1.3, respectively. These figures illustrate the lack of convexity when using small streaming batch sizes $C_\rho$, such as the traditional SG descent and its average estimate $\{C_\rho = 1, \rho = 0\}$, which leads to divergence. It is important to note that the lack of convexity is reflected in the absence of a positive $\mu_\nu$, which can only be counteracted by larger streaming batch sizes $C_\rho$. Moreover, figure 1d excludes the traditional SG descent $\{C_\rho = 1, \rho = 0\}$ due to its lack of

convexity. The figure demonstrates that larger ($C_\rho = 64$) and non-decreasing ($\rho \geq 0$) streaming batches can converge even under challenging settings.

## 4.2 Historical Hourly Weather Data

To illustrate our methodology on real-life time-dependent streaming data, we consider some historical hourly weather data.[5] This dataset comprises approximately five years (roughly 45000 data points) of high temporal resolution hourly measurements encompassing various weather attributes, including temperature, humidity, and air pressure. The dataset encompasses thirty-six cities, resulting in a dimensionality of $d = 36$. In our analysis, we specifically focus on the hourly temperature measurements. To account for monthly and annual seasonality, we preprocess the data by subtracting the corresponding monthly and annual averages. This ensures that the analysis is centered around the deviations from the expected seasonal patterns.

### 4.2.1 Geometric Median

In the presence of noisy data, robust estimators such as the geometric median are preferred. The geometric median, introduced by Haldane (Haldane, 1948), is a robust generalization of the real median. Its efficiency makes it particularly well-suited for handling high-dimensional streaming data (Cardot et al., 2013; Godichon-Baggioni, 2016). To estimate the geometric median of $X \in \mathbb{R}^d$, we minimize the objective function $F(\theta) = \mathbb{E}[\|X - \theta\| - \|X\|]$ using stochastic gradient estimates $\nabla_\theta f(\theta) = (X - \theta)/\|X - \theta\|$. The existence, uniqueness, and robustness (breakdown point) of the geometric median have been studied in Kemperman (1987); Gervini (2008). It is important to note that this objective function only possesses locally strong convexity properties (Cardot et al., 2013). However, by projecting the gradients, it is possible to adapt the proof of optimality in a streaming setting, as demonstrated by Gadat & Panloup (2023). Alternatively, if $X$ is bounded, one can adapt the approach of Cardot et al. (2012) to the streaming setting, which shows that the estimates remain bounded and projection is unnecessary in such cases. In our analysis, we choose not to project our estimates, as doing so would obscure the errors we intend to explore.

### 4.2.2 Discussion

Similar to previous evaluations, we assess the performance using the mean quadratic error of the parameter estimates over one hundred replications, denoted as $\mathbb{E}[\|\theta_{N_t} - \theta^*\|^2]$. In this comparison, we contrast our estimates with the geometric median estimate obtained through Weiszfeld's algorithm (Weiszfeld & Plastria, 2009). We suppose our data are standard Gaussian random variables centered at $(\theta_i)_{1 \leq i \leq d}$, where each $\theta_i$ is randomly selected from the range $[-d, d]$. To align with the reasoning of Cardot et al. (2013), we set $C_\gamma = \sqrt{d}$ and choose $\alpha = 2/3$.

Figure 2 presents the results of the geometric median estimation, following the procedure outlined in section 4.2.1. Despite the robustness of the geometric median, noticeable fluctuations are observed in figure 2, which arise from the time-dependency and noise present in the weather measurements. Figure 2a emphasizes the importance of utilizing a mini-batch size $C_\rho$ to stabilize the optimization process and ensure convexity through larger streaming batches $C_\rho$. However, to achieve reasonable convergence, it is crucial to employ increasing streaming batches with positive streaming rates $\rho > 0$, as illustrated in figures 2b and 2c. These figures demonstrate an enhanced decay of the SSG when the streaming rate $\rho$ is increased. However, the lack of convergence improvement in figure 2c can be attributed to the use of $\beta = 0$, which neglects the potential benefits of leveraging additional observations to accelerate convergence. For further insights into this matter, refer to Godichon-Baggioni et al. (2023). As discussed after Theorem 2, one way to achieve this acceleration is by setting $\alpha = 2/3$ and $\beta = 1/3$. Figure 2d demonstrates that simply selecting $\beta = 1/3$ enables this acceleration. Moreover, $\beta = 1/3$ ensures optimal convergence robust to any streaming rate $\rho$. Selecting an appropriate $\beta > 0$ is particularly crucial when $C_\rho$ is large, as robustness is a fundamental aspect of the geometric median method. Most surprisingly, excellent convergence with a final error as low as $10^{-5}$ can be achieved by combining increasing streaming batches with averaging. This is exemplified in figure 2d with $C_\rho = 64$, $\rho > 0$, and $\beta = 1/3$. Based on these real-life experiments and the discussion in section 4.1.4,

---

[5]The historical hourly weather dataset can be found on `https://www.kaggle.com/datasets/selfishgene/historical-hourly-weather-data`.

Figure 2: Geometric median for various data streams $n_t = C_\rho t^\rho$. See section 4.2 for details.

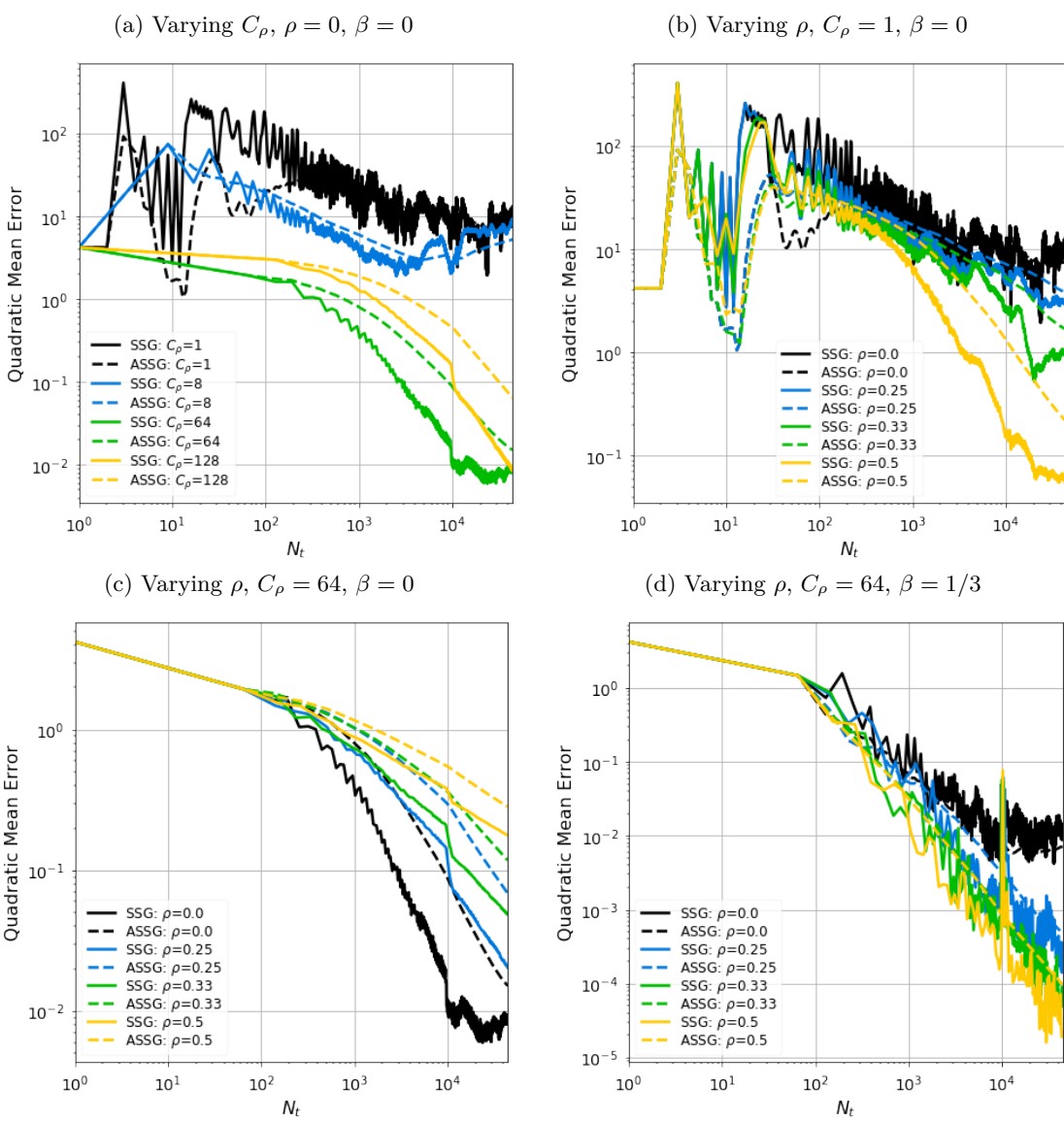

we speculate that the sequence of scores $(\nabla_\theta f_t(\theta^*))$ constitutes a martingale difference sequence that extends beyond our specific examples. Notably, our findings indicate that $\sigma = 1/2$ holds even in the presence of long-range dependence. Thus, the complexity associated with Theorem 2 appears to be an artifact of the proof, which relies on Assumptions 3-p to 5-p with $p = 4$.

## 5    Conclusions and Future Work

In this paper, we explored SGD-based methods in the context of streaming data. We extended the analysis of the unbiased i.i.d. case by Godichon-Baggioni et al. (2023) to include time-dependency and biasedness. By leveraging their insights, we investigated the effectiveness of first-order SO methods in a streaming setting, where the assumption of unbiased i.i.d. samples no longer holds. Our non-asymptotic analysis established novel heuristics that bridge the gap between dependence, biases, and the convexity levels of the SO problem. These heuristics enabled accelerated convergence in complex problems, offering promising opportunities for efficient optimization in streaming settings. Specifically, our findings demonstrated that (i) time-varying

mini-batch SGD methods can break long- and short-range dependence structures, (ii) biased SGD methods can achieve comparable performance to their unbiased counterparts, and (iii) incorporating Polyak-Ruppert averaging can accelerate the convergence. We validated our theoretical findings by conducting a series of experiments using both simulated and real-life time-dependent data.

**Future perspectives.** There are several ways to expand our work about stochastic streaming algorithms: (a) we can extend our analysis to include streaming batches of any size (and not as a function of streaming batch size $C_\rho$ and streaming rates $\rho$), e.g., Godichon-Baggioni et al. (2023) discuss random streaming batches with negative and positive drift. (b) an extension to non-strongly convex objectives could be advantageous as it will provide more insight into how we should choose our learning rates (Bach & Moulines, 2013; Nemirovski et al., 2009; Necoara et al., 2019; Gadat & Panloup, 2023). (c) learning rates should be made adaptive so they are robust to poor initialization and require less tuning; an adaptive learning rate is essential for practitioners as it builds a form of universality across applications, e.g., see Duchi et al. (2011); Kingma & Ba (2014); Godichon-Baggioni & Tarrago (2023). (d) non-parametric analysis could improve our theoretical results for large values of $d$. (e) we have focused on results in quadratic mean but another way to strengthen our non-asymptotic guarantees could be high probability bounds (Durmus et al., 2021; 2022); for any $\delta \in (0, 1)$, we could obtain bounds on the sequence $\{\|\theta_t - \theta^*\| : t \in \mathbb{N}\}$ that holds with probability at least $1 - \delta$.

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

## A Proofs

To begin, we establish recursive relationships for the desired quantities $\delta_t = \mathbb{E}[\|\theta_t - \theta^*\|^2]$ and $\bar{\delta}_t = \mathbb{E}[\|\bar{\theta}_t - \theta^*\|^2]$. These relationships hold for any values of $(\gamma_t)$, $(\nu_t)$, $(\sigma_t)$, and $(n_t)$. Subsequently, we substitute the specific functional forms of these parameters, which lead to the results presented in Theorem 1 and Theorem 2.

Before presenting the proofs, it is important to revisit a recurring argument employed to non-asymptotically bound $\delta_t$ and $\bar{\delta}_t$:

**Proposition 1** (Godichon-Baggioni et al. (2023)). *Suppose $(\omega_t)$, $(\alpha_t)$, $(\eta_t)$, and $(\beta_t)$ to be some non-negative sequences satisfying the recursive relation,*

$$\omega_t \leq [1 - 2\lambda\alpha_t + \eta_t\alpha_t]\omega_{t-1} + \beta_t\alpha_t, \tag{17}$$

*with $\omega_0 \geq 0$ and $\lambda > 0$. Let $C_\omega \geq 1$ be such that $\lambda\alpha_t \leq 1$ for all $t \geq t_\omega$ with $t_\omega = \inf\{t \geq 1 : C_\omega\eta_t \leq \lambda\}$. Then, for $(\alpha_t)$ and $(\eta_t)$ decreasing, we have the upper bound on $(\omega_t)$ given by*

$$\omega_t \leq \tau_t + \frac{1}{\lambda} \max_{t/2 \leq i \leq t} \beta_i, \tag{18}$$

*with $\tau_t = \exp\left(-\lambda \sum_{i=t/2}^{t} \alpha_i\right) \left[\exp\left(C_\omega \sum_{i=1}^{t} \eta_i\alpha_i\right)\left(\omega_0 + \frac{1}{\lambda}\max_{1 \leq i \leq t}\beta_i\right) + \sum_{i=1}^{t/2-1}\beta_i\alpha_i\right]$.*

Proposition 1 presents a straightforward method for bounding $(\omega_t)$ in (17). The resulting bound in (18) comprises a sub-exponential term $\tau_t$ and a noise term $\lambda^{-1} \max_{t/2 \leq i \leq t} \beta_i$. Hence, our primary objective is to minimize the noise term without compromising the inherent decay of the sub-exponential term. It is worth noting that the sub-exponential term $\tau_t$ diminishes exponentially as $t \to \infty$.

During our proofs, specific functions will be introduced, resulting in various generalized harmonic numbers that can be bounded using the integral test for convergence. Additionally, to express our findings in terms of $N_t = \sum_{i=1}^{t} n_i$, we will utilize the inequality $(N_t/2C_\rho)^{1/(1+\rho)} \leq t \leq (2N_t/C_\rho)^{1/(1+\rho)}$, as demonstrated in Godichon-Baggioni et al. (2023). For the sake of simplicity in notation, we will employ the functions $\psi_x(t)$ and $\psi_x^y(t)$, defined as mappings from $\mathbb{R}_+ \setminus 0$ to $\mathbb{R}$, given by

$$\psi_x(t) = \begin{cases} t^{1-x}/(1-x) & \text{if } x < 1, \\ 1 + \log(t) & \text{if } x = 1, \\ x/(x-1) & \text{if } x > 1, \end{cases} \quad \text{and} \quad \psi_x^y(t) = \begin{cases} t^{(1-x)/(1+y)}/(1-x) & \text{if } x < 1, \\ 1 + \log(t^{1/(1+y)}) & \text{if } x = 1, \\ x/(x-1) & \text{if } x > 1, \end{cases} \quad (19)$$

with $y \in \mathbb{R}_+$ such that $\psi_x^y(t) = \psi_x(t^{1/(1+y)})$. Consequently, for any $x \geq 0$, we have $\sum_{i=1}^{t} i^{-x} \leq \psi_x(t)$. Additionally, when considering $\psi_x^y(t)/t$, we find that if $x < 1$, then $\psi_x^y(t)/t = \mathcal{O}(t^{-(x+y)/(1+y)})$. If $x = 1$, it is $\mathcal{O}(\log(t)t^{-1})$. And if $x > 1$, it becomes $\mathcal{O}(t^{-1})$. Therefore, for any $x_0, x_1, x_2, y \geq 0$, we can conclude that $\psi_{x_0}^y(t)/t = \tilde{\mathcal{O}}(t^{-(x_0+y)/(1+y)})$, and $\psi_{x_1}^y(t)\psi_{x_2}^y(t)/t = \tilde{\mathcal{O}}(t^{-(x_1+x_2+y-1)/(1+y)})$, where $\tilde{\mathcal{O}}(\cdot)$ denotes the suppression of logarithmic factors.

In Lemma 1, we establish an explicit recursive relation for $\delta_t$, which provides a non-asymptotic bound on the $t$-th estimate of (7). We achieve this using classical techniques from stochastic approximations (Benveniste et al., 2012; Kushner & Yin, 2003).

*Proof of Lemma 1.* By taking the quadratic norm on (7), expanding it, and taking the expectation, we can derive the equation

$$\delta_t = \delta_{t-1} + \gamma_t^2 \mathbb{E}[\|\nabla_\theta f_t(\theta_{t-1})\|^2] - 2\gamma_t \mathbb{E}[\langle \nabla_\theta f_t(\theta_{t-1}), \theta_{t-1} - \theta^* \rangle], \quad (20)$$

with $\delta_0 \geq 0$. To bound the second term on the right-hand side of (20), we use Assumptions 4-p and 5-p for $p = 2$. This allows us to derive the inequality,

$$\mathbb{E}[\|\nabla_\theta f_t(\theta_{t-1})\|^2] \leq 2\mathbb{E}[\|\nabla_\theta f_t(\theta_{t-1}) - \nabla_\theta f_t(\theta^*)\|^2] + 2\mathbb{E}[\|\nabla_\theta f_t(\theta^*)\|^2] \leq 2C_\kappa^2 \delta_{t-1} + 2\sigma_t^2, \quad (21)$$

using that $\|x+y\|^p \leq 2^{p-1}(\|x\|^p + \|y\|^p)$. As mentioned in Bottou et al. (2018); Nesterov et al. (2018), (2) implies that $\langle \nabla_\theta F(\theta), \theta - \theta^* \rangle \geq \mu \|\theta - \theta^*\|^2$ for all $\theta \in \Theta$. Thus, as $F$ is $\mu$-quasi-strongly convex (2) and $\theta_{t-1}$ is $\mathcal{F}_{t-1}$-measurable (Assumption 3-p), we can bound the third term on the right-hand side of (20) as follows:

$$\mathbb{E}[\langle \nabla_\theta f_t(\theta_{t-1}), \theta_{t-1} - \theta^* \rangle] = \mathbb{E}[\langle \nabla_\theta F(\theta_{t-1}), \theta_{t-1} - \theta^* \rangle] + \mathbb{E}[\langle \mathbb{E}[\nabla_\theta f_t(\theta_{t-1})|\mathcal{F}_{t-1}] - \nabla_\theta F(\theta_{t-1}), \theta_{t-1} - \theta^* \rangle]$$
$$\geq \mu \delta_{t-1} - D_\nu \nu_t \delta_{t-1} - B_\nu \nu_t \delta_{t-1}^{\frac{1}{2}}, \quad (22)$$

since

$$\mathbb{E}[\langle \mathbb{E}[\nabla_\theta f_t(\theta_{t-1})|\mathcal{F}_{t-1}] - \nabla_\theta F(\theta_{t-1}), \theta_{t-1} - \theta^* \rangle] \geq -\mathbb{E}[\|\mathbb{E}[\nabla_\theta f_t(\theta_{t-1})|\mathcal{F}_{t-1}] - \nabla_\theta F(\theta_{t-1})\| \|\theta_{t-1} - \theta^*\|]$$
$$\geq -\sqrt{\mathbb{E}[\|\mathbb{E}[\nabla_\theta f_t(\theta_{t-1})|\mathcal{F}_{t-1}] - \nabla_\theta F(\theta_{t-1})\|^2]}\sqrt{\mathbb{E}[\|\theta_{t-1} - \theta^*\|^2]}$$
$$\geq -\sqrt{\nu_t^2(D_\nu^2 \delta_{t-1} + B_\nu^2)}\sqrt{\delta_{t-1}} \geq -D_\nu \nu_t \delta_{t-1} - B_\nu \nu_t \sqrt{\delta_{t-1}},$$

by Jensen's inequality, Cauchy–Schwarz inequality, Hölder's inequality, and Assumption 3-p with $p = 2$. Hence, by applying the inequalities (21) and (22) to (20), we obtain that

$$\delta_t \leq [1 - 2\mu\gamma_t + 2D_\nu \nu_t \gamma_t + 2C_\kappa^2 \gamma_t^2]\delta_{t-1} + 2B_\nu \nu_t \gamma_t \delta_{t-1}^{\frac{1}{2}} + 2\sigma_t^2 \gamma_t^2$$
$$\leq [1 - (\mu - 2D_\nu \nu_t)\gamma_t + 2C_\kappa^2 \gamma_t^2]\delta_{t-1} + \frac{B_\nu^2}{\mu}\nu_t^2 \gamma_t + 2\sigma_t^2 \gamma_t^2,$$

using Young's inequality[6] in the second line; $2B_\nu \nu_t \gamma_t \delta_{t-1}^{\frac{1}{2}} \leq \mu \gamma_t \delta_{t-1} + B_\nu^2 \nu_t^2 \gamma_t/\mu$. Bounding the projected estimate (8) follows from the property that $\mathbb{E}[\|\mathcal{P}_\Theta(\theta) - \theta\|^2] \leq \mathbb{E}[\|\theta - \theta\|^2]$, for all $\theta \in \mathbb{R}^d$ and $\theta^* \in \Theta$, as mentioned in Zinkevich (2003). □

The following corollary directly follows by applying Proposition 1 to the recursive relation for $\delta_t$ derived in Lemma 1.

**Corollary 1.** *Let $\delta_t = \mathbb{E}[\|\theta_t - \theta^*\|^2]$, where $(\theta_t)$ either follows the recursion in (7) or (8). Suppose Assumptions 1 and 3-p to 5-p hold for $p = 2$. Let $\mathbb{1}_{\{\nu_t = \mathcal{C}\}}$ and $\mathbb{1}_{\{\nu_t \neg \mathcal{C}\}}$ indicate whether $(\nu_t)$ is constant or not. If $\mu_\nu = \mu - \mathbb{1}_{\{\nu_t = \mathcal{C}\}} 2D_\nu \nu_t > 0$, then, for any learning rate $(\gamma_t)$, we have*

$$\delta_t \leq \pi_t + \frac{2B_\nu^2}{\mu\mu_\nu} \max_{t/2 \leq i \leq t} \nu_i^2 + \frac{4}{\mu_\nu} \max_{t/2 \leq i \leq t} \sigma_i^2 \gamma_i,$$

*with*

$$\pi_t = \exp\left(-\frac{\mu_\nu}{2} \sum_{i=t/2}^{t} \gamma_i\right) \left[\exp\left(\mathbb{1}_{\{\nu_t \neg \mathcal{C}\}} 2C_\delta D_\nu \sum_{i=1}^{t} \nu_i \gamma_i\right) \exp\left(2C_\delta C_\kappa^2 \sum_{i=1}^{t} \gamma_i^2\right)\right.$$
$$\left.\left(\delta_0 + \frac{2B_\nu^2}{\mu\mu_\nu} \max_{1 \leq i \leq t} \nu_i^2 + \frac{4}{\mu_\nu} \max_{1 \leq i \leq t} \sigma_i^2 \gamma_i\right) + \frac{B_\nu^2}{\mu} \sum_{i=1}^{t/2-1} \nu_i^2 \gamma_i + 2 \sum_{i=1}^{t/2-1} \sigma_i^2 \gamma_i^2\right].$$

*Proof of Corollary 1.* First, we introduce the indicator function for whether $(\nu_t)$ is constant $(= \mathcal{C})$ or not $(\neg\mathcal{C})$. With this notation, we can rewrite $\delta_t$ from Lemma 1 as follows:

$$\delta_t \leq [1 - (\mu_\nu - \mathbb{1}_{\{\nu_t \neg \mathcal{C}\}} 2D_\nu \nu_t)\gamma_t + 2C_\kappa^2 \gamma_t^2]\delta_{t-1} + \frac{B_\nu^2}{\mu} \nu_t^2 \gamma_t + 2\sigma_t^2 \gamma_t^2, \tag{23}$$

with $\mu_\nu = \mu - \mathbb{1}_{\{\nu_t = \mathcal{C}\}} 2D_\nu \nu_t > 0$. Let us consider $C_\delta = \max\{1, 2C_\kappa^2, (\mu_\nu/2)^2, 2\mathbb{1}_{\{\nu_t \neg \mathcal{C}\}} D_\nu\}$. By doing so, we can rewrite (23) as follows:

$$\delta_t \leq [1 - \mu_\nu \gamma_t + C_\delta (\nu_t + \gamma_t) \gamma_t]\delta_{t-1} + \frac{B_\nu^2}{\mu} \nu_t^2 \gamma_t + 2\sigma_t^2 \gamma_t^2. \tag{24}$$

Let $t_\delta$ denote $\inf\{t : C_\delta (\nu_t + \gamma_t) \leq \mu_\nu/2\}$. Then, for any $t \geq t_\delta$, we have $\gamma_t(\mu_\nu/2)^2 \leq C_\delta \gamma_t \leq \mu_\nu/2$, i.e., $\mu_\nu \gamma_t/2 \leq 1$. Thus, the conditions of Proposition 1 are satisfied. Then, applying Proposition 1 to inequality (24), we can conclude the proof. □

*Proof of Theorem 1.* By substituting the functions $\gamma_t = C_\gamma n_t^\beta t^{-\alpha}$, $\nu_t = n_t^{-\nu}$, $\sigma_t = C_\sigma n_t^{-\sigma}$, and $n_t = \lceil C_\rho t^\rho \rceil \geq C_\rho t^\rho$ into the bound of Corollary 1, we obtain that

$$\delta_t \leq \pi_t + \frac{2^{1+2\rho\nu} B_\nu^2}{\mu\mu_\nu C_\rho^{2\nu} t^{2\rho\nu}} + \frac{2^{2+\rho(2\sigma-\beta)+\alpha} C_\sigma^2 C_\gamma C_\rho^\beta}{\mu_\nu C_\rho^{2\sigma} t^{\rho(2\sigma-\beta)+\alpha}} \tag{25}$$

$$\leq \pi_t + \frac{2^{(2+6\rho\nu)/(1+\rho)} B_\nu^2}{\mu\mu_\nu C_\rho^{2\nu/(1+\rho)} N_t^{2\rho\nu/(1+\rho)}} + \frac{2^{(7+6\rho\sigma)/(1+\rho)} C_\sigma^2 C_\gamma}{\mu_\nu C_\rho^{(2\sigma-\beta-\alpha)/(1+\rho)} N_t^{(\rho(2\sigma-\beta)+\alpha)/(1+\rho)}}, \tag{26}$$

---

[6]If $a, b, c > 0$, $p, q > 1$ such that $1/p + 1/q = 1$, then $ab \leq a^p c^p/p + b^q/qc^q$.

with $\mu_\nu = \mu - \mathbb{1}_{\{\rho=0\}} 2D_\nu C_\rho^{-\nu} > 0$, and

$$
\begin{aligned}
\pi_t &\leq \exp\left(-\frac{\mu_\nu C_\gamma C_\rho^\beta t^{1+\rho\beta-\alpha}}{2^2}\right)\left[\exp\left(\frac{\mathbb{1}_{\{\rho\neq0\}} 2C_\delta D_\nu C_\gamma C_\rho^\beta \psi_{\alpha-\rho(\beta-\nu)}(t)}{C_\rho^\nu}\right)\exp\left(\frac{4(\alpha-\rho\beta)C_\delta C_\kappa^2 C_\gamma^2 C_\rho^{2\beta}}{(2\alpha-2\rho\beta-1)}\right)\right. \\
&\left(\delta_0 + \frac{2B_\nu^2}{\mu\mu_\nu C_\rho^{2\nu}} + \frac{4C_\sigma^2 C_\gamma C_\rho^\beta}{\mu_\nu C_\rho^{2\sigma}}\right) + \frac{B_\nu^2 C_\gamma C_\rho^\beta \psi_{\alpha-\rho(\beta-2\nu)}(t/2)}{\mu C_\rho^{2\nu}} + \frac{4(\alpha-\rho(\beta-\sigma))C_\sigma^2 C_\gamma^2 C_\rho^{2\beta}}{(2\alpha-2\rho(\beta-\sigma)-1)C_\rho^{2\sigma}}\right] \\
&\leq \exp\left(-\frac{\mu C_\gamma N_t^{(1+\rho\beta-\alpha)/(1+\rho)}}{2^{(3+\rho(2+\beta)-\alpha)/(1+\rho)} C_\rho^{(1-\beta-\alpha)/(1+\rho)}}\right)\left[\exp\left(\frac{\mathbb{1}_{\{\rho\neq0\}} 2C_\delta D_\nu C_\gamma C_\rho^\beta \psi_{\alpha-\rho(\beta-\nu)}^\rho(2N_t/C_\rho)}{C_\rho^\nu}\right)\right. \\
&\exp\left(\frac{4(\alpha-\rho\beta)C_\delta C_\kappa^2 C_\gamma^2 C_\rho^{2\beta}}{(2\alpha-2\rho\beta-1)}\right)\left(\delta_0 + \frac{2B_\nu^2}{\mu\mu_\nu C_\rho^{2\nu}} + \frac{4C_\sigma^2 C_\gamma C_\rho^\beta}{\mu_\nu C_\rho^{2\sigma}}\right) \\
&\left. + \frac{B_\nu^2 C_\gamma C_\rho^\beta \psi_{\alpha-\rho(\beta-2\nu)}^\rho(N_t/C_\rho)}{\mu C_\rho^{2\nu}} + \frac{4(\alpha-\rho(\beta-\sigma))C_\sigma^2 C_\gamma^2 C_\rho^{2\beta}}{(2\alpha-2\rho(\beta-\sigma)-1)C_\rho^{2\sigma}}\right], \quad (27)
\end{aligned}
$$

with help of an integral test for convergence[7], the functions $\psi_x(t)$ and $\psi_x^y(t)$ from (19), and by use of $(N_t/2C_\rho)^{1/(1+\rho)} \leq t \leq (2N_t/C_\rho)^{1/(1+\rho)}$. $\qquad\square$

Next, our focus turns to the analysis of the fourth-order rate, denoted as $\Delta_t = \mathbb{E}[\|\theta_t - \theta^*\|^4]$, for the recursive estimates given by equations (7) and (8). Similar to the approach in Corollary 1, we commence a comprehensive investigation of the general case with $(\gamma_t)$, $(\nu_t)$, $(\sigma_t)$, and $(n_t)$.

**Lemma 2.** Let $\Delta_t = \mathbb{E}[\|\theta_t - \theta^*\|^4]$, where $(\theta_t)$ either follows the recursion in (7) or (8). Suppose Assumptions 1 and 3-p to 5-p hold for $p = 4$. Let $\mathbb{1}_{\{\nu_t=\mathcal{C}\}}$ and $\mathbb{1}_{\{\nu_t\neg\mathcal{C}\}}$ indicate whether $(\nu_t)$ is constant or not. If $\mu_\nu' = \mu - \mathbb{1}_{\{\nu_t=\mathcal{C}\}} 2D_\nu^4 \nu_t^4/\mu^3 > 0$, then for any learning rate $(\gamma_t)$, we have

$$
\Delta_t \leq \Pi_t + \frac{4B_\nu^4}{\mu^3\mu_\nu'} \max_{t/2\leq i\leq t} \nu_i^4 + \frac{1024}{\mu\mu_\nu'} \max_{t/2\leq i\leq t} \sigma_i^4\gamma_i^2 + \frac{96}{\mu_\nu'} \max_{t/2\leq i\leq t} \sigma_i^4\gamma_i^3,
$$

with $\Pi_t$ given as

$$
\begin{aligned}
&\exp\left(-\frac{\mu_\nu'}{4}\sum_{i=t/2}^t \gamma_i\right)\left[\exp\left(\frac{\mathbb{1}_{\{\nu_t\neg\mathcal{C}\}} C_\Delta D_\nu^4}{\mu^3}\sum_{i=1}^t \nu_i^4\gamma_i\right)\exp\left(\frac{256C_\Delta C_\kappa^4}{\mu}\sum_{i=1}^t \gamma_i^3\right)\exp\left(24C_\Delta C_\kappa^4\sum_{i=1}^t \gamma_i^4\right)\right. \\
&\left(\Delta_0 + \frac{4B_\nu^4}{\mu^3\mu_\nu'}\max_{1\leq i\leq t}\nu_i^4 + \frac{1024}{\mu\mu_\nu'}\max_{1\leq i\leq t}\sigma_i^4\gamma_i^2 + \frac{96}{\mu_\nu'}\max_{1\leq i\leq t}\sigma_i^4\gamma_i^3\right) + \frac{B_\nu^4}{\mu^3}\sum_{i=1}^{t/2-1}\nu_i^4\gamma_i + \frac{256}{\mu}\sum_{i=1}^{t/2-1}\sigma_i^4\gamma_i^3 + 24\sum_{i=1}^{t/2-1}\sigma_i^4\gamma_i^4\right].
\end{aligned}
$$

*Proof of Lemma 2.* The derivation of the recursive step sequence for the fourth-order moment, denoted by $\Delta_t$, in (7), follows a similar methodology as for the second-order moment in Corollary 1. By applying the same approach used to derive (20), we can take the quadratic norm of (7), expand the norm, square both sides of the equation, and take the conditional expectation on both sides. This leads us to the following expression:

$$
\begin{aligned}
\Delta_t =&\Delta_{t-1} + \gamma_t^4\mathbb{E}[\|\nabla_\theta f_t(\theta_{t-1})\|^4] + 4\gamma_t^2\mathbb{E}[\langle\nabla_\theta f_t(\theta_{t-1}),\theta_{t-1}-\theta^*\rangle^2] + 2\gamma_t^2\mathbb{E}[\|\theta_{t-1}-\theta^*\|^2\|\nabla_\theta f_t(\theta_{t-1})\|^2] \\
&- 4\gamma_t\mathbb{E}[\|\theta_{t-1}-\theta^*\|^2\langle\nabla_\theta f_t(\theta_{t-1}),\theta_{t-1}-\theta^*\rangle] - 4\gamma_t^3\mathbb{E}[\|\nabla_\theta f_t(\theta_{t-1})\|^2\langle\nabla_\theta f_t(\theta_{t-1}),\theta_{t-1}-\theta^*\rangle] \\
\leq&\Delta_{t-1} + \gamma_t^4\mathbb{E}[\|\nabla_\theta f_t(\theta_{t-1})\|^4] + 6\gamma_t^2\mathbb{E}[\|\theta_{t-1}-\theta^*\|^2\|\nabla_\theta f_t(\theta_{t-1})\|^2] \\
&- 4\gamma_t\mathbb{E}[\|\theta_{t-1}-\theta^*\|^2\langle\nabla_\theta f_t(\theta_{t-1}),\theta_{t-1}-\theta^*\rangle] + 4\gamma_t^3\mathbb{E}[\|\theta_{t-1}-\theta^*\|\|\nabla_\theta f_t(\theta_{t-1})\|^3],
\end{aligned}
$$

where we have made use of Cauchy-Schwarz inequality. Next, we can utilize Young's inequality to simplify the terms. By applying Young's inequality, we have: $4\gamma_t^3\|\theta_{t-1}-\theta^*\|\|\nabla_\theta f_t(\theta_{t-1})\|^3 \leq$

---

[7] $\sum_{i=1}^t i^{2\rho(\beta-\sigma)-2\alpha} \leq (2\alpha-2\rho(\beta-\sigma))/(2\alpha-2\rho(\beta-\sigma)-1)$ as $\nu > 0$, $\sigma \in [0,1/2]$, $\rho \in [0,1)$, $\beta \in [0,1]$, and $\alpha-\rho\beta \in (1/2,1)$.

$2\gamma_t^4\|\nabla_\theta f_t(\theta_{t-1})\|^4 + 2\gamma_t^2\|\theta_{t-1}-\theta^*\|^2\|\nabla_\theta f_t(\theta_{t-1})\|^2$ and $8\gamma_t^2\|\theta_{t-1}-\theta^*\|^2\|\nabla_\theta f_t(\theta_{t-1})\|^2 \leq (\mu\gamma_t/2)\|\theta_{t-1}-\theta^*\|^4 + 32\mu^{-1}\gamma_t^3\|\nabla_\theta f_t(\theta_{t-1})\|^4$. These inequalities allow us to obtain the simplified expression:

$$\Delta_t \leq [1+\mu\gamma_t/2]\Delta_{t-1} + 3\gamma_t^4\mathbb{E}[\|\nabla_\theta f_t(\theta_{t-1})\|^4] + 32\mu^{-1}\gamma_t^3\mathbb{E}[\|\nabla_\theta f_t(\theta_{t-1})\|^4]$$
$$- 4\gamma_t\mathbb{E}[\|\theta_{t-1}-\theta^*\|^2\langle\nabla_\theta f_t(\theta_{t-1}),\theta_{t-1}-\theta^*\rangle].$$

In order to bound the fourth-order term $\mathbb{E}[\|\nabla_\theta f_t(\theta_{t-1})\|^4]$, we can utilize several assumptions. Firstly, we make use of the Lipschitz continuity of $\nabla_\theta f_t$ (as stated in Assumption 4-p). Additionally, we consider the bounds on $\nabla_\theta f_t(\theta^*)$ given in Assumption 5-p, and the fact that $\theta_{t-1}$ is $\mathcal{F}_{t-1}$-measurable (as stated in Assumption 3-p). Combining these assumptions, we can show that $\mathbb{E}[\|\nabla_\theta f_t(\theta_{t-1})\|^4] \leq 8C_\kappa^4\Delta_{t-1} + 8\sigma_t^4$. Thus,

$$\Delta_t \leq [1+\mu\gamma_t/2 + 256\mu^{-1}C_\kappa^4\gamma_t^3 + 24C_\kappa^4\gamma_t^4]\Delta_{t-1} + 256\mu^{-1}\sigma_t^4\gamma_t^3 + 24\sigma_t^4\gamma_t^4$$
$$- 4\gamma_t\mathbb{E}[\|\theta_{t-1}-\theta^*\|^2\langle\nabla_\theta f_t(\theta_{t-1}),\theta_{t-1}-\theta^*\rangle]. \tag{28}$$

Next, by employing similar arguments as in the proof of Corollary 1, along with Young's inequality and Assumption 3-p (with $p=4$), we have that

$$4\gamma_t\mathbb{E}[\|\theta_{t-1}-\theta^*\|^2\langle\mathbb{E}[\nabla_\theta f_t(\theta_{t-1})|\mathcal{F}_{t-1}]-\nabla_\theta F(\theta_{t-1}),\theta_{t-1}-\theta^*\rangle]$$
$$\geq -4\gamma_t\mathbb{E}[\|\theta_{t-1}-\theta^*\|^3\|\mathbb{E}[\nabla_\theta f_t(\theta_{t-1})|\mathcal{F}_{t-1}]-\nabla_\theta F(\theta_{t-1})\|]$$
$$\geq -3\mu\gamma_t\Delta_{t-1} - \mu^{-3}\gamma_t\mathbb{E}[\|\mathbb{E}[\nabla_\theta f_t(\theta_{t-1})|\mathcal{F}_{t-1}]-\nabla_\theta F(\theta_{t-1})\|^4]$$
$$\geq -3\mu\gamma_t\Delta_{t-1} - \mu^{-3}\gamma_t D_\nu^4\nu_t^4\Delta_{t-1} - \mu^{-3}\gamma_t B_\nu^4\nu_t^4,$$

Hence, utilizing this inequality, we can bound the last term of (28) as follows,

$$4\gamma_t\mathbb{E}[\|\theta_{t-1}-\theta^*\|^2\langle\nabla_\theta f_t(\theta_{t-1}),\theta_{t-1}-\theta^*\rangle] = 4\gamma_t\mathbb{E}[\|\theta_{t-1}-\theta^*\|^2\langle\mathbb{E}[\nabla_\theta f_t(\theta_{t-1})|\mathcal{F}_{t-1}],\theta_{t-1}-\theta^*\rangle]$$
$$= 4\gamma_t\mathbb{E}[\|\theta_{t-1}-\theta^*\|^2\langle\nabla_\theta F(\theta_{t-1}),\theta_{t-1}-\theta^*\rangle] + 4\gamma_t\mathbb{E}[\|\theta_{t-1}-\theta^*\|^2\langle\mathbb{E}[\nabla_\theta f_t(\theta_{t-1})|\mathcal{F}_{t-1}]-\nabla_\theta F(\theta_{t-1}),\theta_{t-1}-\theta^*\rangle]$$
$$\geq \mu\gamma_t\Delta_{t-1} - \mu^{-3}\gamma_t D_\nu^4\nu_t^4\Delta_{t-1} - \mu^{-3}\gamma_t B_\nu^4\nu_t^4.$$

To summarize, by inserting this into (28) and incorporating the indicator function that determines whether $(\nu_t)$ is constant $(=\mathcal{C})$ or not $(\neg\mathcal{C})$, we obtain the following inequality:

$$\Delta_t \leq \left[1 - \left(\frac{\mu_\nu}{2} - \frac{\mathbb{1}_{\{\nu_t\neg\mathcal{C}\}}D_\nu^4\nu_t^4}{\mu^3}\right)\gamma_t + \frac{256C_\kappa^4\gamma_t^3}{\mu} + 24C_\kappa^4\gamma_t^4\right]\Delta_{t-1} + \frac{B_\nu^4\nu_t^4\gamma_t}{\mu^3} + \frac{256\sigma_t^4\gamma_t^3}{\mu} + 24\sigma_t^4\gamma_t^4, \tag{29}$$

with $\mu'_\nu = \mu - \mathbb{1}_{\{\nu_t=\mathcal{C}\}}2D_\nu^4\nu_t^4/\mu^3 > 0$. Note that $\mu_\nu$ from Corollary 1 is lower bounded by $\mu'_\nu$, and it is strictly lower bounded when $(\nu_t)$ is constant, i.e., $\mu_\nu > \mu'_\nu > 0$. Let $C_\Delta \geq 1$ satisfy the conditions of Proposition 1. The constant $C_\Delta$ is chosen such that $C_\Delta(\mathbb{1}_{\{\nu_t\neg\mathcal{C}\}}D_\nu^4\nu_t^4/\mu^3 + 256C_\kappa^4\gamma_t^2/\mu + 24C_\kappa^4\gamma_t^3) \leq \mu'_\nu/2$, which implies $\mu'_\nu\gamma_t/2 \leq 1$. This condition is possible since the sequence $(\nu_t)$ is non-increasing and $(\gamma_t)$ is decreasing. At last, by applying Proposition 1 to (29), we obtain the desired bound for $\Delta_t$. $\qquad\square$

**Corollary 2.** *Let* $\Delta_t = \mathbb{E}[\|\theta_t-\theta^*\|^4]$, *where* $(\theta_t)$ *either follows the recursion in (7) or (8). Suppose Assumptions 1 and 3-p to 5-p hold for* $p = 4$. *If* $\mu'_\nu = \mu - \mathbb{1}_{\{\rho=0\}}2D_\nu^4/\mu^3C_\rho^{4\nu} > 0$, *then for* $\alpha - \rho\beta \in (1/2,1)$, *we have*

$$\Delta_t \leq \Pi_t + \frac{2^{2+4\rho\nu}B_\nu^4}{\mu^3\mu'_\nu C_\rho^{4\nu}t^{4\rho\nu}} + \frac{2^{2\rho(2\sigma-\beta)+2\alpha}(2^{10}\mu^{-1}+2^7C_\gamma C_\rho^\beta)C_\sigma^4 C_\gamma^2 C_\rho^{2\beta}}{\mu'_\nu C_\rho^{4\sigma}t^{2\rho(2\sigma-\beta)+2\alpha}}, \tag{30}$$

*with* $\Pi_t$ *given in (31) such that* $\Pi_t = \mathcal{O}(\exp(-N_t^{(1+\rho\beta-\alpha)/(1+\rho)}))$.

*Proof of Corollary 2.* By substituting the functions $\gamma_t = C_\gamma n_t^\beta t^{-\alpha}$, $\nu_t = n_t^{-\nu}$, $\sigma_t = C_\sigma n_t^{-\sigma}$, and $n_t = C_\rho t^\rho$ into the bound of Lemma 2, and utilizing the inequality $\gamma_t^3 \leq C_\gamma C_\rho^\beta\gamma_t^2$ due to $\alpha - \rho\beta \in (1/2,1)$, we obtain

(30). In this inequality, we have $\mu'_\nu = \mu - \mathbb{1}_{\{\rho=0\}} 2 D_\nu^4 / \mu^3 C_\rho^{4\nu} > 0$. Furthermore, $\Pi_t$ can be bounded as follows:

$$
\begin{aligned}
\Pi_t \leq{} & \exp\left(-\frac{\mu'_\nu C_\gamma C_\rho^\beta}{4} \sum_{i=t/2}^t i^{\rho\beta-\alpha}\right) \Bigg[ \exp\left(\frac{\mathbb{1}_{\{\rho\neq0\}} C_\Delta D_\nu^4 C_\gamma C_\rho^\beta}{\mu^3 C_\rho^{4\nu}} \sum_{i=1}^t i^{\rho(\beta-4\nu)-\alpha}\right) \\
& \exp\left(\frac{2^8 C_\Delta C_\kappa^4 C_\gamma^3 C_\rho^{3\beta}}{\mu} \sum_{i=1}^t i^{3\rho\beta-3\alpha}\right) \exp\left(24 C_\Delta C_\kappa^4 C_\gamma^4 C_\rho^{4\beta} \sum_{i=1}^t i^{4\rho\beta-4\alpha}\right) \\
& \left(\Delta_0 + \frac{4 B_\nu^4}{\mu^3 \mu'_\nu C_\rho^{4\nu}} + \frac{1024 C_\sigma^4 C_\gamma^2 C_\rho^{2\beta}}{\mu \mu'_\nu C_\rho^{4\sigma}} + \frac{96 C_\sigma^4 C_\gamma^3 C_\rho^{3\beta}}{\mu'_\nu C_\rho^{4\sigma}}\right) + \frac{B_\nu^4 C_\gamma C_\rho^\beta}{\mu^3 C_\rho^{4\nu}} \sum_{i=1}^{t/2-1} i^{\rho(\beta-4\nu)-\alpha} \\
& + \frac{256 C_\sigma^4 C_\gamma^3 C_\rho^{3\beta}}{\mu C_\rho^{4\sigma}} \sum_{i=1}^{t/2-1} i^{\rho(3\beta-4\sigma)-3\alpha} + \frac{24 C_\sigma^4 C_\gamma^4 C_\rho^{4\beta}}{C_\rho^{4\sigma}} \sum_{i=1}^{t/2-1} i^{4\rho(\beta-\sigma)-4\alpha} \Bigg] \\
\leq{} & \exp\left(-\frac{\mu'_\nu C_\gamma C_\rho^\beta t^{1+\rho\beta-\alpha}}{2^3}\right) \Bigg[ \exp\left(\frac{\mathbb{1}_{\{\rho\neq0\}} C_\Delta D_\nu^4 C_\gamma C_\rho^\beta \psi_{\alpha-\rho(\beta-4\nu)}^0(t)}{\mu^3 C_\rho^{4\nu}}\right) \exp\left(\frac{2^{10} C_\Delta C_\kappa^4 C_\gamma^3 C_\rho^{3\beta}}{\mu}\right) \\
& \exp\left(2^6 C_\Delta C_\kappa^4 C_\gamma^4 C_\rho^{4\beta}\right) \left(\Delta_0 + \frac{2^2 B_\nu^4}{\mu^3 \mu'_\nu C_\rho^{4\nu}} + \frac{2^{10} C_\sigma^4 C_\gamma^2 C_\rho^{2\beta}}{\mu \mu'_\nu C_\rho^{4\sigma}} + \frac{2^7 C_\sigma^4 C_\gamma^3 C_\rho^{3\beta}}{\mu'_\nu C_\rho^{4\sigma}}\right) \\
& + \frac{B_\nu^4 C_\gamma C_\rho^\beta \psi_{\alpha-\rho(\beta-4\nu)}^0(t/2)}{\mu^3 C_\rho^{4\nu}} + \frac{2^{10} C_\sigma^4 C_\gamma^3 C_\rho^{3\beta}}{\mu C_\rho^{4\sigma}} + \frac{2^6 C_\sigma^4 C_\gamma^4 C_\rho^{4\beta}}{C_\rho^{4\sigma}} \Bigg],
\end{aligned}
\tag{31}
$$

with help of the integral test for convergence.[8]  $\qquad\square$

**Lemma 3.** *Let $\bar\delta_t = \mathbb{E}[\|\bar\theta_t - \theta^*\|^2]$ with $\bar\theta_n$ given by (9), where $(\theta_t)$ either follows the recursion in (7) or (8). Suppose Assumptions 1 to 6 hold for $p = 4$. In addition, Assumption 7 must hold true if $(\theta_t)$ follows the recursion in (8), which is indicated by $\mathbb{1}_{\{D_\Theta<\infty\}}$. Then, for any learning rate $(\gamma_t)$, we have*

$$
\begin{aligned}
\bar\delta_t^{1/2} \leq{} & \frac{\Lambda^{1/2}}{N_t}\left(\sum_{i=1}^t n_i^{2(1-\sigma)}\right)^{1/2} + \frac{C_\sigma'^{1/2}}{\mu N_t}\left(\sum_{i=1}^t n_i^{2(1-\sigma-\sigma')}\right)^{1/2} + \frac{2^{1/2} B_\nu^{1/2}}{\mu N_t}\left(\sum_{j=2}^t \left(n_j \nu_j \sum_{i=1}^{j-1} n_i \sigma_i\right)\right)^{1/2} \\
& + \frac{1}{\mu N_t}\sum_{i=1}^{t-1} \delta_i^{1/2}\left|\frac{n_{i+1}}{\gamma_{i+1}} - \frac{n_i}{\gamma_i}\right| + \frac{n_t}{\mu \gamma_t N_t}\delta_t^{1/2} + \frac{n_1}{\mu N_t}\left(\frac{1}{\gamma_1} + 2^{1/2}(C_\nabla + C_\kappa)\right)\delta_0^{1/2} \\
& + \frac{2^{1/2}(C_\nabla^2 + C_\kappa^2)^{1/2}}{\mu N_t}\left(\sum_{i=1}^{t-1} n_{i+1}^2 \delta_i\right)^{1/2} + \frac{C_\nabla''}{\mu N_t}\sum_{i=0}^{t-1} n_{i+1}\Delta_i^{1/2}, \\
& + \frac{2^{3/4}(C_\nabla^2 + C_\kappa^2)^{1/2}}{\mu N_t}\left(\sum_{j=1}^{t-1}\left((D_\nu \delta_j^{1/2} + 2^{1/2} B_\nu) n_{j+1} \nu_{j+1} \sum_{i=0}^{j-1} n_{i+1}\delta_i^{1/2}\right)\right)^{1/2},
\end{aligned}
$$

*with $\Lambda = \mathrm{Tr}(\nabla_\theta^2 F(\theta^*)^{-1} \Sigma \nabla_\theta^2 F(\theta^*)^{-1})$ and $C_\nabla'' = C_\nabla'/2 + \mathbb{1}_{\{D_\Theta<\infty\}} 2 G_\Theta / D_\Theta^2$.*

*Proof of Lemma 3.* The proof is presented in two parts. In the first part, we consider the case where $(\theta_t)$ follows the recursion given by (7). In the second part, we examine the case of (8). Let's assume that $(\theta_t)$ is obtained using the recursion in (7). To proceed, we apply the observation made by Polyak & Juditsky (1992), which observe that

$$
\begin{aligned}
\nabla_\theta^2 F(\theta^*)(\theta_{t-1} - \theta^*) ={} & -\nabla_\theta f_t(\theta^*) + \nabla_\theta f_t(\theta_{t-1}) - [\nabla_\theta f_t(\theta_{t-1}) - \nabla_\theta f_t(\theta^*) - \nabla_\theta F(\theta_{t-1})] \\
& - [\nabla_\theta F(\theta_{t-1}) - \nabla_\theta^2 F(\theta^*)(\theta_{t-1} - \theta^*)],
\end{aligned}
$$

---

[8] $\sum_{i=1}^t i^{3\rho\beta-3\alpha} \leq 3 < 2^2$ and $\sum_{i=1}^t i^{4\rho(\beta-x)-4\alpha} \leq 2$ for any $x \geq 0$ as $\alpha - \rho\beta \in (1/2, 1)$.

where $\nabla_\theta^2 F(\theta^*)$ is invertible with lowest eigenvalue greater than $\mu$, i.e., $\nabla_\theta^2 F(\theta^*) \geq \mu \mathbb{I}_d$. By summing the individual parts, taking the quadratic norm and expectation, and applying Minkowski's inequality, we obtain the following inequality:

$$
\left(\mathbb{E}\left[\left\|\bar{\theta}_t - \theta^*\right\|^2\right]\right)^{\frac{1}{2}} \leq \left(\mathbb{E}\left[\left\|\nabla_\theta^2 F(\theta^*)^{-1} \frac{1}{N_t} \sum_{i=1}^t n_i \nabla_\theta f_i(\theta^*)\right\|^2\right]\right)^{\frac{1}{2}}
$$

$$
+ \left(\mathbb{E}\left[\left\|\nabla_\theta^2 F(\theta^*)^{-1} \frac{1}{N_t} \sum_{i=1}^t n_i \nabla_\theta f_i(\theta_{i-1})\right\|^2\right]\right)^{\frac{1}{2}}
$$

$$
+ \left(\mathbb{E}\left[\left\|\nabla_\theta^2 F(\theta^*)^{-1} \frac{1}{N_t} \sum_{i=1}^t n_i \left[\nabla_\theta f_i(\theta_{i-1}) - \nabla_\theta f_i(\theta^*) - \nabla_\theta F(\theta_{i-1})\right]\right\|^2\right]\right)^{\frac{1}{2}}
$$

$$
+ \left(\mathbb{E}\left[\left\|\nabla_\theta^2 F(\theta^*)^{-1} \frac{1}{N_t} \sum_{i=1}^t n_i \left[\nabla_\theta F(\theta_{i-1}) - \nabla_\theta^2 F(\theta^*)(\theta_{i-1} - \theta^*)\right]\right\|^2\right]\right)^{\frac{1}{2}}. \quad (32)
$$

First term of (32): As $(\nabla_\theta f_t(\theta^*))$ is a square-integrable sequences on $\mathbb{R}^d$ (Assumption 3-p), we have

$$
\mathbb{E}\left[\left\|\nabla_\theta^2 F(\theta^*)^{-1} \frac{1}{N_t} \sum_{i=1}^t n_i \nabla_\theta f_i(\theta^*)\right\|^2\right] = \frac{1}{N_t^2} \sum_{i=1}^t n_i^2 \mathbb{E}\left[\left\|\nabla_\theta^2 F(\theta^*)^{-1} \nabla_\theta f_i(\theta^*)\right\|^2\right]
$$

$$
+ \frac{2}{N_t^2} \sum_{1 \leq i < j \leq t} n_i n_j \mathbb{E}\left[\left\langle \nabla_\theta^2 F(\theta^*)^{-1} \nabla_\theta f_i(\theta^*), \nabla_\theta^2 F(\theta^*)^{-1} \nabla_\theta f_j(\theta^*)\right\rangle\right],
$$

Here, the first term can be bounded using Assumption 6, as follows:

$$
\frac{1}{N_t^2} \sum_{i=1}^t n_i^2 \mathbb{E}\left[\left\|\nabla_\theta^2 F(\theta^*)^{-1} \nabla_\theta f_i(\theta^*)\right\|^2\right] \leq \frac{\Lambda}{N_t^2} \sum_{i=1}^t n_i^{2(1-\sigma)} + \frac{C_\sigma'}{\mu^2 N_t^2} \sum_{i=1}^t n_i^{2(1-\sigma-\sigma')},
$$

where $\Lambda$ denotes $\mathrm{Tr}[\nabla_\theta^2 F(\theta^*)^{-1} \Sigma \nabla_\theta^2 F(\theta^*)^{-1}]$. For the second term, we use Cauchy-Schwarz inequality, Hölder's inequality, and Assumptions 3-p and 5-p to show that

$$
\frac{2}{N_t^2} \sum_{1 \leq i < j \leq t} n_i n_j \mathbb{E}\left[\left\langle \nabla_\theta^2 F(\theta^*)^{-1} \nabla_\theta f_i(\theta^*), \nabla_\theta^2 F(\theta^*)^{-1} \nabla_\theta f_j(\theta^*)\right\rangle\right]
$$

$$
\leq \frac{2}{\mu^2 N_t^2} \sum_{1 \leq i < j \leq t} n_i n_j \mathbb{E}\left[\left\langle \nabla_\theta f_i(\theta^*), \nabla_\theta f_j(\theta^*) - \nabla_\theta F(\theta^*)\right\rangle\right]
$$

$$
\leq \frac{2}{\mu^2 N_t^2} \sum_{1 \leq i < j \leq t} n_i n_j \mathbb{E}\left[\left\|\nabla_\theta f_i(\theta^*)\right\| \left\|[\mathbb{E}[\nabla_\theta f_j(\theta^*)|\mathcal{F}_{j-1}] - \nabla_\theta F(\theta^*)]\right\|\right]
$$

$$
\leq \frac{2}{\mu^2 N_t^2} \sum_{1 \leq i < j \leq t} n_i n_j \sqrt{\mathbb{E}\left[\left\|\nabla_\theta f_i(\theta^*)\right\|^2\right]} \sqrt{\mathbb{E}\left[\left\|[\mathbb{E}[\nabla_\theta f_j(\theta^*)|\mathcal{F}_{j-1}] - \nabla_\theta F(\theta^*)]\right\|^2\right]}
$$

$$
\leq \frac{2 B_\nu}{\mu^2 N_t^2} \sum_{1 \leq i < j \leq t} n_i n_j \sigma_i \nu_j = \frac{2 B_\nu}{\mu^2 N_t^2} \sum_{j=2}^t \left(n_j \nu_j \sum_{i=1}^{j-1} n_i \sigma_i\right).
$$

Thus, combining these finding gives us

$$\left(\mathbb{E}\left[\left\|\nabla_\theta^2 F\left(\theta^*\right)^{-1}\frac{1}{N_t}\sum_{i=1}^t n_i\nabla_\theta f_i\left(\theta^*\right)\right\|^2\right]\right)^{\frac{1}{2}}\le\frac{\Lambda^{\frac{1}{2}}}{N_t}\left(\sum_{i=1}^t n_i^{2(1-\sigma)}\right)^{\frac{1}{2}}$$

$$+\frac{C_\sigma'^{1/2}}{\mu N_t^{1/2}}\left(\sum_{i=1}^t n_i^{2(1-\sigma-\sigma')}\right)^{\frac{1}{2}}+\frac{2^{1/2}B_\nu^{1/2}}{\mu N_t}\left(\sum_{j=2}^t\left(n_j\nu_j\sum_{i=1}^{j-1}n_i\sigma_i\right)\right)^{\frac{1}{2}}.\tag{33}$$

Second term of (32): First, we use that $\frac{1}{N_t}\sum_{i=1}^t n_i\nabla_\theta f_i(\theta_{i-1})=\frac{1}{N_t}\sum_{i=1}^t\frac{n_i}{\gamma_i}(\theta_{i-1}-\theta_i)=\frac{1}{N_t}\sum_{i=1}^{t-1}(\theta_i-\theta^*)(\frac{n_{i+1}}{\gamma_{i+1}}-\frac{n_i}{\gamma_i})-\frac{1}{N_t}(\theta_t-\theta^*)\frac{n_t}{\gamma_t}+\frac{1}{N_t}(\theta_0-\theta^*)\frac{n_1}{\gamma_1}$, which leads to an upper bound on normed quantity $\|\nabla_\theta^2 F(\theta^*)^{-1}\frac{1}{N_t}\sum_{i=1}^t n_i\nabla_\theta f_i(\theta_{i-1})\|$ given by

$$\frac{1}{\mu N_t}\sum_{i=1}^{t-1}\|\theta_i-\theta^*\|\left|\frac{n_{i+1}}{\gamma_{i+1}}-\frac{n_i}{\gamma_i}\right|+\frac{1}{\mu N_t}\|\theta_t-\theta^*\|\frac{n_t}{\gamma_t}+\frac{1}{\mu N_t}\|\theta_0-\theta^*\|\frac{n_1}{\gamma_1}.$$

Rewriting this in terms of $\delta_t=\mathbb{E}[\|\theta_t-\theta^*\|^2]$, gives us a bound for second term of (32);

$$\frac{1}{\mu N_t}\sum_{i=1}^{t-1}\delta_i^{\frac{1}{2}}\left|\frac{n_{i+1}}{\gamma_{i+1}}-\frac{n_i}{\gamma_i}\right|+\frac{n_t}{\mu\gamma_t N_t}\delta_t^{\frac{1}{2}}+\frac{n_1}{\mu\gamma_1 N_t}\delta_0^{\frac{1}{2}}.\tag{34}$$

Third term of (32): Here, we use that $\mathbb{E}[\|\nabla_\theta^2 F(\theta^*)^{-1}\frac{1}{N_t}\sum_{i=1}^t n_i[\nabla_\theta f_i(\theta_{i-1})-\nabla_\theta f_i(\theta^*)-\nabla_\theta F(\theta_{i-1})]\|^2]$ can be derived as

$$\frac{1}{\mu^2 N_t^2}\left[\sum_{i=1}^t n_i^2\mathbb{E}[\|\nabla_\theta f_i(\theta_{i-1})-\nabla_\theta f_i(\theta^*)-\nabla_\theta F(\theta_{i-1})\|^2]\right.$$

$$\left.+2\sum_{i<j}^t n_i n_j\mathbb{E}[\langle\nabla_\theta f_i(\theta_{i-1})-\nabla_\theta f_i(\theta^*)-\nabla_\theta F(\theta_{i-1}),\nabla_\theta f_j(\theta_{j-1})-\nabla_\theta f_j(\theta^*)-\nabla_\theta F(\theta_{j-1})\rangle]\right].$$

Next, we use Cauchy-Schwarz inequality, Assumption 4-p and (3) to show that

$$\sum_{i=1}^t n_i^2\mathbb{E}[\|\nabla_\theta f_i(\theta_{i-1})-\nabla_\theta f_i(\theta^*)-\nabla_\theta F(\theta_{i-1})\|^2]\le 2(C_\nabla^2+C_\kappa^2)\sum_{i=1}^t n_i^2\delta_{i-1}.$$

Similarly, for the other term, we note that

$$\mathbb{E}[\langle\nabla_\theta f_i(\theta_{i-1})-\nabla_\theta f_i(\theta^*)-\nabla_\theta F(\theta_{i-1}),\nabla_\theta f_j(\theta_{j-1})-\nabla_\theta f_j(\theta^*)-\nabla_\theta F(\theta_{j-1})\rangle]$$

$$\le\sqrt{\mathbb{E}[\|\nabla_\theta f_i(\theta_{i-1})-\nabla_\theta f_i(\theta^*)-[\nabla_\theta F(\theta_{i-1})-\nabla_\theta F(\theta^*)]\|^2]}$$

$$\sqrt{\mathbb{E}[\|\mathbb{E}[\nabla_\theta f_j(\theta_{j-1})|\mathcal{F}_{j-1}]-\nabla_\theta F(\theta_{j-1})-[\mathbb{E}[\nabla_\theta f_j(\theta^*)|\mathcal{F}_{j-1}]-\nabla_\theta F(\theta^*)]\|^2]}$$

$$\le\sqrt{2\mathbb{E}[\|\nabla_\theta f_i(\theta_{i-1})-\nabla_\theta f_i(\theta^*)\|^2]+2\mathbb{E}[\|\nabla_\theta F(\theta_{i-1})-\nabla_\theta F(\theta^*)\|^2]}$$

$$\sqrt{2\mathbb{E}[\|\mathbb{E}[\nabla_\theta f_j(\theta_{j-1})|\mathcal{F}_{j-1}]-\nabla_\theta F(\theta_{j-1})\|^2]+2\mathbb{E}[\|\mathbb{E}[\nabla_\theta f_j(\theta^*)|\mathcal{F}_{j-1}]-\nabla_\theta F(\theta^*)\|^2]}$$

$$\le\sqrt{2(C_\kappa^2+C_\nabla^2)\delta_{i-1}}\sqrt{2D_\nu^2\nu_j^2\delta_{j-1}+4B_\nu^2\nu_j^2}$$

$$\le 2^{1/2}(C_\kappa^2+C_\nabla^2)^{1/2}\delta_{i-1}^{1/2}(D_\nu\nu_j\delta_{j-1}^{1/2}+2^{1/2}B_\nu\nu_j),$$

using $\mathcal{F}_{i-1} \subset \mathcal{F}_{j-1}$ since $i < j$, Cauchy–Schwarz inequality, Hölder's inequality, $\|a+b\|^p \leq 2^{p-1}(\|a\|^p + \|b\|^p)$ with $p \in \mathbb{N}$, Assumptions 3-p and 4-p, and (3). Thus, the third term of (32) can be upper bounded by

$$\frac{2^{1/2}(C_\kappa^2 + C_\nabla^2)^{1/2}}{\mu N_t} \left( \sum_{i=1}^t n_i^2 \delta_{i-1} \right)^{1/2} + \frac{2^{3/4}(C_\nabla^2 + C_\kappa^2)^{1/2}}{\mu N_t} \left( \sum_{j=2}^t \left( (D_\nu \delta_{j-1}^{1/2} + 2^{1/2} B_\nu) n_j \nu_j \sum_{i=1}^{j-1} n_i \delta_{i-1}^{1/2} \right) \right)^{1/2}.$$
(35)

Fourth term of (32): Here, we use that (4) implies $\forall \theta$, $\|\nabla_\theta F(\theta) - \nabla_\theta^2 F(\theta^*)(\theta - \theta^*)\| \leq C_\nabla' \|\theta - \theta^*\|^2/2$ (Nesterov et al., 2018), which gives the upper bound $\frac{C_\nabla'}{2\mu N_t} \sum_{i=1}^t n_i \Delta_{i-1}^{1/2}$ using the definition $\Delta_t = \mathbb{E}[\|\theta_t - \theta^*\|^4]$. Combining the terms (33) to (35) into (32), together with shifting the indices and collecting the $\delta_0$ terms, gives use the desired bound for when $(\theta_t)$ follows (7).

Now, assume that $(\theta_t)$ is derived from the recursion in (8). As above, we follow the steps of Polyak & Juditsky (1992), in which, we can rewrite (8) to $\frac{1}{\gamma_t}(\theta_{t-1} - \theta_t) = \nabla_\theta f_t(\theta_{t-1}) - \frac{1}{\gamma_t}\Omega_t$, where $\Omega_t = \mathcal{P}_\Theta(\theta_{t-1} - \gamma_t \nabla_\theta f_t(\theta_{t-1})) - (\theta_{t-1} - \gamma_t \nabla_\theta f_t(\theta_{t-1}))$. Thus, summing the parts, taking the norm and expectation, and using the Minkowski's inequality, yields the same terms as in (32), but with an additional term regarding $\Omega_t$, namely

$$\left( \mathbb{E}\left[ \left\| \nabla_\theta^2 F(\theta^*)^{-1} \frac{1}{N_t} \sum_{i=1}^t \frac{n_i}{\gamma_i} \Omega_i \right\|^2 \right] \right)^{\frac{1}{2}} \leq \frac{1}{\mu N_t} \sum_{i=1}^t \frac{n_i}{\gamma_i} \sqrt{\mathbb{E}\left[ \|\Omega_i\|^2 \mathbb{1}_{\{\theta_{i-1} - \gamma_i \nabla_\theta f_i(\theta_{i-1}) \notin \Theta\}} \right]},$$
(36)

using (Godichon-Baggioni, 2016, Lemma 4.3). Next, we note that

$$\|\Omega_t\|^2 = \|\mathcal{P}_\Theta(\theta_{t-1} - \gamma_t \nabla_\theta f_t(\theta_{t-1})) - \theta_{t-1} + \gamma_t \nabla_\theta f_t(\theta_{t-1})\|^2$$
$$\leq 2\|\mathcal{P}_\Theta(\theta_{t-1} - \gamma_t \nabla_\theta f_t(\theta_{t-1})) - \theta_{t-1}\|^2 + 2\gamma_t^2 \|\nabla_\theta f_t(\theta_{t-1})\|^2$$
$$= 2\|\mathcal{P}_\Theta(\theta_{t-1} - \gamma_t \nabla_\theta f_t(\theta_{t-1})) - \mathcal{P}_\Theta(\theta_{t-1})\|^2 + 2\gamma_t^2 \|\nabla_\theta f_t(\theta_{t-1})\|^2$$
$$\leq 2\|\theta_{t-1} - \gamma_t \nabla_\theta f_t(\theta_{t-1}) - \theta_{t-1}\|^2 + 2\gamma_t^2 \|\nabla_\theta f_t(\theta_{t-1})\|^2 \leq 4\gamma_t^2 G_\Theta^2,$$

as $\mathcal{P}_\Theta$ is Lipschitz and $\|\nabla_\theta f_t(\theta)\|^2 \leq G_\Theta^2$ for any $\theta \in \Theta$. This means that the inner expectation of (36), $\mathbb{E}[\|\Omega_t\|^2 \mathbb{1}_{\{\theta_{t-1} - \gamma_t \nabla_\theta f_t(\theta_{t-1}) \notin \Theta\}}] = 4\gamma_t^2 G_\Theta^2 \mathbb{P}[\theta_{t-1} - \gamma_t \nabla_\theta f_t(\theta_{t-1}) \notin \Theta]$. Moreover, as in (Godichon-Baggioni & Portier, 2017, Theorem 4.2) with use of Lemma 2, we know that $\mathbb{P}[\theta_{t-1} - \gamma_t \nabla_\theta f_t(\theta_{t-1}) \notin \Theta] \leq \Delta_t/D_\Theta^4$, where $D_\Theta = \inf_{\theta \in \partial \Theta} \|\theta - \theta^*\|$ with $\partial \Theta$ denoting the frontier of $\Theta$. Thus, (36) can then be bounded by

$$\frac{1}{\mu N_t} \sum_{i=1}^t \frac{n_i}{\gamma_i} \sqrt{\mathbb{E}\left[ \|\Omega_i\|^2 \mathbb{1}_{\{\theta_{i-1} - \gamma_i \nabla_\theta f_i(\theta_{i-1}) \notin \Theta\}} \right]} \leq \frac{2 G_\Theta}{\mu D_\Theta^2 N_t} \sum_{i=1}^t n_{i+1} \Delta_i^{1/2},$$

since the sequence $(n_t)$ is either constant or increasing, meaning $\forall t, n_t/n_{t+1} \leq 1$. At last, let $C_\nabla'' = C_\nabla'/2 + \mathbb{1}_{\{D_\Theta < \infty\}} 2 G_\Theta/D_\Theta^2$ indicate whether $(\theta_t)$ follows (8) or not. $\square$

*Proof of Theorem 2.* This result can be obtained by simplifying and bounding each term of Lemma 3, using the bounds provided by Theorem 1 and Lemma 2. By inserting the functions $\gamma_t = C_\gamma n_t^\beta t^{-\alpha}$, $\nu_t = n_t^{-\nu}$,

$\sigma_t = C_\sigma n_t^{-\sigma}$, and $n_t = C_\rho t^\rho$ into the bound of Lemma 3, we obtain that

$$
\begin{aligned}
\bar{\delta}_t^{1/2} \le{} & \frac{\Lambda^{1/2}}{N_t^{1/2}} \mathbb{1}_{\{\sigma=1/2\}} + \frac{\Lambda^{1/2}C_\rho^{1-\sigma}}{N_t}\left(\sum_{i=1}^{t} i^{2\rho(1-\sigma)}\right)^{1/2} \mathbb{1}_{\{\sigma\neq 1/2\}} + \frac{C_\sigma'^{1/2}C_\rho^{1-\sigma-\sigma'}}{\mu N_t}\left(\sum_{i=1}^{t} i^{2\rho(1-\sigma-\sigma')}\right)^{1/2} \\
& + \frac{(\rho(1-\beta)+\alpha)C_\rho}{\mu C_\gamma C_\rho^\beta N_t}\sum_{i=1}^{t-1} i^{\rho(1-\beta)+\alpha-1}\delta_i^{1/2} + \frac{2^{1/2}B_\nu^{1/2}C_\sigma^{1/2}C_\rho}{\mu C_\rho^{(\sigma+\nu)/2}N_t}\left(\sum_{j=2}^{t}\left(j^{\rho(1-\nu)}\sum_{i=1}^{j-1} i^{\rho(1-\sigma)}\right)\right)^{1/2} \\
& + \frac{C_\rho t^{\rho(1-\beta)+\alpha}}{\mu C_\gamma C_\rho^\beta N_t}\delta_t^{1/2} + \frac{C_\rho}{\mu N_t}\left(\frac{1}{C_\gamma C_\rho^\beta}+2^{1/2}\left(C_\kappa+C_\nabla\right)\right)\delta_0^{1/2} + \frac{2^{1/2+\rho}(C_\kappa^2+C_\nabla^2)^{1/2}C_\rho}{\mu N_t}\left(\sum_{i=1}^{t-1} i^{2\rho}\delta_i\right)^{1/2} \\
& + \frac{2^\rho C_\nabla'' C_\rho}{\mu N_t}\sum_{i=0}^{t-1} i^\rho \Delta_i^{1/2} + \frac{2^{3/4+\rho(2-\nu)/2}(C_\nabla^2+C_\kappa^2)^{1/2}C_\rho}{\mu C_\rho^{\nu/2}N_t}\left(\sum_{j=1}^{t-1}\left((D_\nu\delta_j^{1/2}+2^{1/2}B_\nu)j^{\rho(1-\nu)}\sum_{i=1}^{j-1} i^\rho\delta_i^{1/2}\right)\right)^{1/2},
\end{aligned}
$$

using $n_{i+1}/n_i \le 2^\rho$ and that $|n_{i+1}/\gamma_{i+1}-n_i/\gamma_i| \le (\rho(1-\beta)+\alpha)C_\rho^{1-\beta}/C_\gamma i^{1-\rho(1-\beta)-\alpha}$ as $\rho(1-\beta)+\alpha \le 1-\rho$ with $\rho \in [0,1)$. Next, as $\sigma \in [0,1/2]$ and $\sigma' \in (0,1/2]$, we have $\sum_{i=1}^{t} i^{2\rho(1-\sigma-\sigma')} \le t^{1+2\rho(1-\sigma-\sigma')}/(1+2\rho(1-\sigma-\sigma'))$, where $t \le (2N_t/C_\rho)^{1/(1+\rho)}$. Similarly, as $\nu \in (0,\infty)$, we have that

$$
\begin{aligned}
\sum_{j=2}^{t-1}\left(j^{\rho(1-\nu)}\sum_{i=1}^{j-1} i^{\rho(1-\sigma)}\right) &\le \sum_{j=1}^{t-1} j^{\rho(1-\nu)}\sum_{i=1}^{t-1} i^{\rho(1-\sigma)} \le \psi_{\rho(\nu-1)}(t)\psi_{\rho(\sigma-1)}(t) \\
&\le \psi_{\rho(\nu-1)}^\rho(2N_t/C_\rho)\psi_{\rho(\sigma-1)}^\rho(2N_t/C_\rho),
\end{aligned}
$$

using the $\psi$-function defined in (19). Hence, $\sqrt{\psi_{\rho(\sigma-1)}^\rho(2N_t/C_\rho)\psi_{\rho(\nu-1)}^\rho(2N_t/C_\rho)}/N_t$ is $\tilde{\mathcal{O}}(N_t^{-\rho(\sigma+\nu)/2(1+\rho)})$. From (25) we know that $\delta_t \le D_\delta/t^\delta$ with

$$
D_\delta = \sup_{t\in\mathbb{N}} \pi_t t^\delta + \frac{2^{1+2\rho\nu}B_\nu^2}{\mu\mu_\nu C_\rho^{2\nu}} + \frac{2^{2+\rho(2\sigma-\beta)+\alpha}C_\sigma^2 C_\gamma C_\rho^\beta}{\mu_\nu C_\rho^{2\sigma}},
$$

and $\delta = \mathbb{1}_{\{B_\nu=0\}}(\rho(2\sigma-\beta)+\alpha) + \mathbb{1}_{\{B_\nu\neq 0\}}\min\{\rho(2\sigma-\beta)+\alpha, 2\rho\nu\}$, yielding

$$
\begin{aligned}
\sum_{j=1}^{t-1}\left((D_\nu\delta_j^{1/2}+2^{1/2}B_\nu)j^{\rho(1-\nu)}\sum_{i=1}^{j-1} i^\rho\delta_i^{1/2}\right) &\le D_\delta^{1/2}\sum_{j=1}^{t-1}\left((D_\nu D_\delta^{1/2}j^{-\delta/2}+2^{1/2}B_\nu)j^{\rho(1-\nu)}\psi_{\delta/2-\rho}(t)\right) \\
&\le D_\nu D_\delta\psi_{\delta/2-\rho}(t)\psi_{\delta/2+\rho(\nu-1)}(t) + 2^{1/2}B_\nu D_\delta^{1/2}\psi_{\delta/2-\rho}(t)\psi_{\rho(\nu-1)}(t) \\
&\le D_\nu D_\delta\psi_{\delta/2-\rho}^\rho(2N_t/C_\rho)\psi_{\delta/2+\rho(\nu-1)}^\rho(2N_t/C_\rho) + 2^{1/2}B_\nu D_\delta^{1/2}\psi_{\delta/2-\rho}^\rho(2N_t/C_\rho)\psi_{\rho(\nu-1)}^\rho(2N_t/C_\rho),
\end{aligned}
$$

if $\delta/2 - \rho \ge 0$. Hence, $\sqrt{\psi_{\delta/2-\rho}^\rho(2N_t/C_\rho)\psi_{\delta/2+\rho(\nu-1)}^\rho(2N_t/C_\rho)}/N_t$ is $\tilde{\mathcal{O}}(N_t^{-(\delta+\rho\nu)/2(1+\rho)})$, and $\sqrt{\psi_{\delta/2-\rho}^\rho(2N_t/C_\rho)\psi_{\rho(\nu-1)}^\rho(2N_t/C_\rho)}/N_t$ is $\tilde{\mathcal{O}}(N_t^{-(\delta/2+\rho\nu)/2(1+\rho)})$. Next, we define $\bar{\pi}_t = \sum_{i=1}^{t} i^2\pi_i \ge \sum_{i=1}^{t}\pi_i$ such that $\pi_t \le t^{-1}\sum_{i=1}^{t}\pi_i \le t^{-1}\bar{\pi}_t \le t^{-1}\bar{\pi}_\infty$ since $\pi_t$ is decreasing. Similarly, let $\bar{\Pi}_t = \sum_{i=1}^{t} i^\rho\Pi_i$. Both $\bar{\pi}_t$ and $\bar{\Pi}_t$ convergences to some finite constant depending on the model's parameters. With use of these notions,

we have

$$
\begin{aligned}
\bar{\delta}_t^{1/2} \leq\ & \frac{\Lambda^{1/2}}{N_t^{1/2}}\mathbb{1}_{\{\sigma=1/2\}} + \frac{2^{1/2}\Lambda^{1/2}C_\rho^{(1-2\sigma)/2(1+\rho)}}{N_t^{(1+2\rho\sigma)/2(1+\rho)}}\mathbb{1}_{\{\sigma\neq 1/2\}} + \frac{2^{1/2}C_\sigma'^{1/2}C_\rho^{(1-2(\sigma+\sigma'))/2(1+\rho)}}{\mu N_t^{(1+2\rho(\sigma+\sigma'))/2(1+\rho)}} \\
& + \frac{2^{2+(7+2\rho(1+\sigma))/2(1+\rho)}C_\sigma C_\rho^{(2-2\sigma-\beta-\alpha)/2(1+\rho)}}{\mu\mu_\nu^{1/2}C_\gamma^{1/2}N_t^{(2+\rho(\beta+2\sigma)-\alpha)/2(1+\rho)}} + \frac{\Gamma C_\rho}{\mu N_t} + \frac{2^{(2+\rho)/(1+\rho)}C_\rho^{(2+\beta-\alpha)/(1+\rho)}\bar{\pi}_\infty}{\mu C_\gamma N_t^{(2+\rho\beta-\alpha)/(1+\rho)}} \\
& + \frac{2^{(1+\rho(1+2\sigma-\beta)+\alpha)/(1+\rho)}(2^5\mu^{-1/2}+2^4C_\gamma^{1/2}C_\rho^{\beta/2})C_\nabla''C_\sigma^2 C_\gamma}{\mu\sqrt{\mu_\nu'}C_\rho^{(1-2\rho\sigma-\alpha)/(1+\rho)}N_t^{(\rho(2\sigma-\beta)+\alpha)/(1+\rho)}} + \mathbb{1}_{\{B_\nu\neq 0\}}\Psi_t \\
& + \frac{2^{(5/2+\rho(5-2\sigma))/2(1+\rho)}D_\nabla^\kappa C_\sigma C_\gamma^{1/2}C_\rho^{(1+\beta-2\sigma+\alpha)/2(1+\rho)}}{\mu\mu_\nu^{1/2}N_t^{(1+\rho(2\sigma-\beta)+\alpha)/(2(1+\rho))}} \\
& + \frac{2^{3/4+\rho(2-\nu)/2}\sqrt{D_\nabla^\kappa}D_\nu^{1/2}D_\delta^{1/2}C_\rho\sqrt{\psi_{\delta/2-\rho}^\rho(2N_t/C_\rho)\psi_{\delta/2+\rho(\nu-1)}^\rho(2N_t/C_\rho)}}{\mu C_\rho^{\nu/2}N_t},
\end{aligned}
$$

as $\alpha-\rho\beta\in(1/2,1)$, where $\mu_\nu'=\mu-\mathbb{1}_{\{\rho=0\}}2D_\nu^4/\mu^3 C_\rho^{4\nu}$, $D_\nabla^\kappa=C_\nabla+C_\kappa$, $C_\nabla''=C_\nabla'+\mathbb{1}_{\{D_\Theta<\infty\}}2G_\Theta/D_\Theta^2$, $\Gamma=2\bar{\pi}_\infty/C_\gamma C_\rho^\beta+(1/C_\gamma C_\rho^\beta+2^{1/2}D_\nabla^\kappa)\delta_0^{1/2}+2^{1/2+\rho}D_\nabla^\kappa\bar{\pi}_\infty^{1/2}+2^\rho C_\nabla''\bar{\Pi}_\infty$, $\delta=\mathbb{1}_{\{B_\nu=0\}}(\rho(2\sigma-\beta)+\alpha)+\mathbb{1}_{\{B_\nu\neq 0\}}\min\{\rho(2\sigma-\beta)+\alpha,2\rho\nu\}$, and $\Psi_t$ given as

$$
\begin{aligned}
& \frac{2^{1/2}B_\nu^{1/2}C_\sigma^{1/2}C_\rho\sqrt{\psi_{\rho(\sigma-1)}^\rho(2N_t/C_\rho)\psi_{\rho(\nu-1)}^\rho(2N_t/C_\rho)}}{\mu C_\rho^{(\sigma+\nu)/2}N_t} + \frac{2^{3(1+\rho\nu)}B_\nu C_\rho^{(1-\beta-\nu-\alpha)/(1+\rho)}}{\mu^{3/2}\mu_\nu^{1/2}C_\gamma N_t^{(1+\rho(\beta+\nu)-\alpha)/(1+\rho)}} \\
& + \frac{2^{1+\rho(2-\nu)/2}B_\nu^{1/2}\sqrt{D_\nabla^\kappa}D_\delta^{1/4}C_\rho\sqrt{\psi_{\delta/2-\rho}^\rho(2N_t/C_\rho)\psi_{\rho(\nu-1)}^\rho(2N_t/C_\rho)}}{\mu C_\rho^{\nu/2}N_t} \\
& + \frac{2^{2(1+\rho\nu)}B_\nu^2 C_\nabla''C_\rho\psi_{\rho(2\nu-1)}^\rho(2N_t/C_\rho)}{\mu^{5/2}\sqrt{\mu_\nu'}C_\rho^{2\nu}N_t} + \frac{2^{3/2+\rho(1+\nu)}B_\nu D_\nabla^\kappa C_\rho\sqrt{\psi_{2\rho(\nu-1)}^\rho(2N_t/C_\rho)}}{\mu^{3/2}\mu_\nu^{1/2}C_\rho^\nu N_t} \\
& + \frac{2^{3/2+\rho\nu}B_\nu C_\rho\psi_{1+\rho(\beta+\nu-1)-\alpha}^\rho(2N_t/C_\rho)}{\mu^{3/2}\mu_\nu^{1/2}C_\gamma C_\rho^{\beta+\nu}N_t},
\end{aligned}
$$

Furthermore, using the $\tilde{\mathcal{O}}$-notation one can show that

$$
\begin{aligned}
\bar{\delta}_t^{1/2} \leq\ & \frac{\Lambda^{1/2}}{N_t^{1/2}}\mathbb{1}_{\{\sigma=1/2\}} + \frac{2^{1/2}\Lambda^{1/2}C_\rho^{(1-2\sigma)/2(1+\rho)}}{N_t^{(1+2\rho\sigma)/2(1+\rho)}}\mathbb{1}_{\{\sigma\neq 1/2\}} + \frac{2^{1/2}C_\sigma'^{1/2}C_\rho^{(1-2(\sigma+\sigma'))/2(1+\rho)}}{\mu N_t^{(1+2\rho(\sigma+\sigma'))/2(1+\rho)}} \\
& + \frac{2^6 C_\sigma C_\rho^{(2-2\sigma-\beta-\alpha)/2(1+\rho)}}{\mu\mu_\nu^{1/2}C_\gamma^{1/2}N_t^{(2+\rho(\beta+2\sigma)-\alpha)/2(1+\rho)}} + \frac{2^7(\mu^{-1/2}+C_\gamma^{1/2}C_\rho^{\beta/2})C_\nabla''C_\sigma^2 C_\gamma}{\mu\sqrt{\mu_\nu'}C_\rho^{(1-2\rho\sigma-\alpha)/(1+\rho)}N_t^{(\rho(2\sigma-\beta)+\alpha)/(1+\rho)}} \\
& + \frac{2^2 D_\nabla^\kappa C_\sigma C_\gamma^{1/2}C_\rho^{(1+\beta-2\sigma+\alpha)/2(1+\rho)}}{\mu\mu_\nu^{1/2}N_t^{(1+\rho(2\sigma-\beta)+\alpha)/(2(1+\rho))}} + \frac{\Gamma C_\rho}{\mu N_t} + \frac{2^2 C_\rho^{(2+\beta-\alpha)/(1+\rho)}\bar{\pi}_\infty}{\mu C_\gamma N_t^{(2+\rho\beta-\alpha)/(1+\rho)}} \\
& + \tilde{\mathcal{O}}(N_t^{-(\delta+\rho\nu)/2(1+\rho)}) + \mathbb{1}_{\{B_\nu\neq 0\}}\Psi_t, \hspace{2cm} (37)
\end{aligned}
$$

where $\Psi_t = \tilde{\mathcal{O}}(N_t^{-\rho(\sigma+\nu)/2(1+\rho)})+\tilde{\mathcal{O}}(N_t^{-(1+\rho(\beta+\nu)-\alpha)/(1+\rho)})+\tilde{\mathcal{O}}(N_t^{-(1+2\rho\nu)/2(1+\rho)})+\tilde{\mathcal{O}}(N_t^{-(\delta/2+\rho\nu)/2(1+\rho)})+\tilde{\mathcal{O}}(N_t^{-2\rho\nu/(1+\rho)})$, implying that $\nu>1/2$ to obtain the desired rate $\bar{\delta}_t=\mathcal{O}(N^{-1})$ if $B_\nu=0$. $\hspace{1cm}\square$

## B  Verifications of Assumptions 3-p and 5-p for the AR model

**Well-specified case.** Consider the well-specified case, in which, we estimate an AR(1) model from the underlying stationary AR(1) process $X_s = \theta^* X_{s-1}+\epsilon_s$ with $|\theta^*|<1$. The squared loss function $f_t(\theta)=$

$n_t^{-1} \sum_{i=1}^{n_t} (X_{N_{t-1}+i} - \theta X_{N_{t-1}+i-1})^2$ with gradient $\nabla_\theta f_t(\theta) = -2n_t^{-1} \sum_{i=1}^{n_t} X_{N_{t-1}+i-1}(X_{N_{t-1}+i} - \theta X_{N_{t-1}+i-1})$. Thus, the objective function is

$$F(\theta) = \mathbb{E}\left[\frac{1}{n_t}\sum_{i=1}^{n_t}(X_{N_{t-1}+i} - \theta X_{N_{t-1}+i-1})^2\right] = \frac{\sigma_\epsilon^2(\theta^* - \theta)^2}{1 - (\theta^*)^2} + \sigma_\epsilon^2,$$

using $\mathbb{E}[X_s] = 0$ and $\mathbb{E}[X_s^2] = \sigma_\epsilon^2/(1 - (\theta^*)^2)$, yielding $\nabla_\theta F(\theta) = 2\sigma_\epsilon^2(\theta - \theta^*)/(1 - (\theta^*)^2)$. Next, to verify Assumption 3-p for $p = 2$, we first note that

$$\mathbb{E}[\nabla_\theta f_t(\theta)|\mathcal{F}_{t-1}] = \frac{2\theta}{n_t}\sum_{i=1}^{n_t}\mathbb{E}\left[X_{N_{t-1}+i-1}^2\Big|\mathcal{F}_{t-1}\right] - \frac{2}{n_t}\sum_{i=1}^{n_t}\mathbb{E}\left[X_{N_{t-1}+i-1}X_{N_{t-1}+i}\Big|\mathcal{F}_{t-1}\right]$$

$$= \frac{2(\theta - \theta^*)}{n_t}\sum_{i=1}^{n_t}\mathbb{E}\left[X_{N_{t-1}+i-1}^2\Big|\mathcal{F}_{t-1}\right] - \frac{2}{n_t}\sum_{i=1}^{n_t}\mathbb{E}\left[X_{N_{t-1}+i-1}\epsilon_{N_{t-1}+i}\Big|\mathcal{F}_{t-1}\right], \qquad (38)$$

as $X_{N_{t-1}+i} = \theta^* X_{N_{t-1}+i-1} + \epsilon_{N_{t-1}+i}$. For the first term of (38), we use that $\mathbb{E}[X_{s+i}|\mathcal{F}_s] = (\theta^*)^i X_s$ and $\mathrm{Var}[X_{s+i}|\mathcal{F}_s] = \sigma_\epsilon^2(1 - (\theta^*)^{2i})/(1 - (\theta^*)^2)$, yielding

$$\sum_{i=1}^{n_t}\mathbb{E}[X_{N_{t-1}+i-1}^2|\mathcal{F}_{t-1}] = X_{N_{t-1}}^2\sum_{i=1}^{n_t}(\theta^*)^{2(i-1)} - \frac{\sigma_\epsilon^2}{(1 - (\theta^*)^2)}\sum_{i=1}^{n_t}(\theta^*)^{2(i-1)} + \frac{\sigma_\epsilon^2 n_t}{1 - (\theta^*)^2}$$

$$= \frac{(1 - (\theta^*)^{2n_t})X_{N_{t-1}}^2}{(1 - (\theta^*)^2)} - \frac{(1 - (\theta^*)^{2n_t})\sigma_\epsilon^2}{(1 - (\theta^*)^2)^2} + \frac{\sigma_\epsilon^2 n_t}{1 - (\theta^*)^2}.$$

Next, the second term of (38) is zero by utilising that $(\epsilon_s)$ is a Martingale difference sequence, i.e., $\mathbb{E}[\epsilon_{s+i}|\mathcal{F}_s] = 0$ and $\mathbb{E}[\epsilon_{s+i}\epsilon_{s+j}|\mathcal{F}_s] = 0$ for $i \neq j$. Thus,

$$\mathbb{E}[\|\mathbb{E}[\nabla_\theta f_t(\theta)|\mathcal{F}_{t-1}] - \nabla_\theta F(\theta)\|^2] = \frac{4(\theta - \theta^*)^2(1 - (\theta^*)^{2n_t})^2\sigma_\epsilon^2}{(1 - (\theta^*)^2)^4 n_t^2}\left(\sigma_\epsilon^2 + \frac{1}{1 - (\theta^*)^2}\right),$$

meaning that Assumption 3-p is verified for $p = 2$ if $(X_s)$ has bounded moments; this is fulfilled by the natural constraint that $|\theta^*| < 1$. Thus, we can deduce that $D_\nu > 0$, $B_\nu = 0$, and $\nu_t$ is $\mathcal{O}(n_t^{-1})$. The remaining assumption can be verified in the same way, in particular, Assumption 5-p is satisfied with $\sigma_t$ is $\mathcal{O}(n_t^{-1/2})$, Assumption 2 with $C_\nabla = 2\sigma_\epsilon^2/(1 - (\theta^*)^2)$ and $C'_\nabla = 0$, and Assumption 6 with $\Sigma = 4\sigma_\epsilon^4/(1 - (\theta^*)^2)$ and $\Sigma_t = 0$. Furthermore, for an AR(1) process $X_s$ constructed using the noise process $\epsilon_s = \sqrt{G_s(H)}z_s$ with Hurst index $H \geq 1/2$, one can verify that $\nu_t^4$ and $\sigma_t^4$ is $\mathcal{O}(n_t^{H-1})$ in Assumptions 3-p and 5-p using the self-similarty property (Nourdin, 2012).

**Misspecified case.** Next, assume that the underlying data generating process follows the MA(1)-process, $X_s = \epsilon_s + \phi^*\epsilon_{s-1}$, with $\phi^* \in \mathbb{R}$. The misspecification error of fitting an AR(1) model to a MA(1) process can be found by minimizing

$$F(\theta) = \mathbb{E}[(X_s - \theta X_{s-1})^2] = \mathbb{E}[(\epsilon_s + \phi^*\epsilon_{s-1} - \theta(\epsilon_{s-1} + \phi^*\epsilon_{s-2}))^2]$$

$$= \mathbb{E}[(\epsilon_s + (\phi^* - \theta)\epsilon_{s-1} - \theta\phi^*\epsilon_{s-2})^2] = \sigma_\epsilon^2(1 + (\phi^* - \theta)^2 + \theta^2(\phi^*)^2),$$

where $\nabla_\theta F(\theta) = 2(\theta - \phi^*)\sigma_\epsilon^2 + 2\theta(\phi^*)^2\sigma_\epsilon^2$. Thus, as $\theta^* = \arg\min_\theta F(\theta) \equiv \arg\min_\theta(\phi^* - \theta)^2 + \theta^2(\phi^*)^2$ is a strictly convex function in $\theta$, we have $\nabla_\theta F(\theta) = 0 \Leftrightarrow 2(\theta - \phi^*) + 2\theta(\phi^*)^2 = 0 \Leftrightarrow 2\theta(1 + (\phi^*)^2) = 2\phi^* \Leftrightarrow \theta = \phi^*/(1 + (\phi^*)^2)$. This means for any $\phi^* \in \mathbb{R}$ then $\theta \in (-1/2, 1/2)$. With this in mind, we can conduct our study of fitting an AR(1) model to the MA(1) process with $\phi^*$ drawn randomly from $\mathbb{R}$ (figure 1b). Furthermore, this reparametrization trick can be used to verify Assumption 3-p: first, we can reparameterize $\nabla_\theta F(\theta) = 2\sigma_\epsilon^2(\theta - \theta^*)(1 + (\phi^*)^2)$ using $\theta^* = \phi^*/(1 + (\phi^*)^2)$. Next, for $\mathbb{E}[\nabla_\theta f_t(\theta)|\mathcal{F}_{t-1}]$ one have that

$$\mathbb{E}[\nabla_\theta f_t(\theta)|\mathcal{F}_{t-1}] = \frac{2\theta}{n_t}\sum_{i=1}^{n_t}\mathbb{E}[X_{N_{t-1}+i-1}^2|\mathcal{F}_{t-1}] - \frac{2}{n_t}\sum_{i=1}^{n_t}\mathbb{E}[X_{N_{t-1}+i-1}X_{N_{t-1}+i}|\mathcal{F}_{t-1}],$$

where

$$\sum_{i=1}^{n_t} \mathbb{E}[X^2_{N_{t-1}+i-1}|\mathcal{F}_{t-1}] = X^2_{N_{t-1}} + \mathbb{E}[X^2_{N_{t-1}+1}|\mathcal{F}_{t-1}] + \cdots + \mathbb{E}[X^2_{N_{t-1}+n_t-1}|\mathcal{F}_{t-1}]$$

$$= X^2_{N_{t-1}} + \sigma^2_\epsilon + (\phi^*)^2\epsilon^2_{N_{t-1}} + \cdots + \sigma^2_\epsilon + (\phi^*)^2\sigma^2_\epsilon$$

$$= X^2_{N_{t-1}} + (\phi^*)^2\epsilon^2_{N_{t-1}} + \sigma^2_\epsilon(n_t - 1) + (\phi^*)^2\sigma^2_\epsilon(n_t - 2)$$

$$= X^2_{N_{t-1}} + (\phi^*)^2(\epsilon^2_{N_{t-1}} - \sigma^2_\epsilon) + (1 + (\phi^*)^2)\sigma^2_\epsilon(n_t - 1),$$

and

$$\sum_{i=1}^{n_t} \mathbb{E}[X_{N_{t-1}+i-1}X_{N_{t-1}+i}|\mathcal{F}_{t-1}] = \phi^* X_{N_{t-1}}\epsilon_{N_{t-1}} + \phi^*\sigma^2_\epsilon(n_t - 1)$$

$$= \theta^*(1 + (\phi^*)^2)X_{N_{t-1}}\epsilon_{N_{t-1}} + \theta^*(1 + (\phi^*)^2)\sigma^2_\epsilon(n_t - 1),$$

using the same white noise properties as for the well-specified case above. This yields,

$$\mathbb{E}[\|\mathbb{E}[\nabla_\theta f_t(\theta)|\mathcal{F}_{t-1}] - \nabla_\theta F(\theta)\|^2] = \frac{4(\theta - \theta^*)^2}{n_t^2}f_{\phi^*}(\epsilon_{N_{t-1}}),$$

where $f_{\phi^*}(\epsilon_{N_{t-1}})$ is finite function depending on the moments of $(\epsilon_{N_{t-1}})$ and $\phi^*$. Hence, we have $D_\nu > 0$ and $B_\nu = 0$ with $\nu_t$ being $\mathcal{O}(n_t^{-1})$. Similarly, it can be verified that $\sigma_t$ are $\mathcal{O}(n_t^{-1/2})$ by use of the reparametrization trick.

