# OpenReview forum: "Learning from time-dependent streaming data with online stochastic algorithms"
_TMLR — Accepted by TMLR_

### Review · Reviewer_gnir · 2023-04-24

**Summary Of Contributions:**

This paper presents several new analysis results for the convergence of stochastic (gradient) algorithms, especially it focuses on the case with time-dependent streaming data via the new conditions proposed in Assumption 1-p to 3-p.

**Audience:**

Yes

**Claims And Evidence:**

Yes

**Requested Changes:**

1. The proof of Lemma 1 leads to a condition that depends on whether $\upsilon_t$ is constant or not, while it seems that the same condition is changed to $\rho = 0$ or not in the presentation of Theorem 1. In addition to the lack of justification presented in the paper, the reviewer notices that $\rho=0$ only implies that $n_t$ will be a constant, which  doesn't necessarily imply that $\upsilon_t$ is constant. Again, the paper shall provide a clear justification about this.

2. It is confusing that paper mention about "conserving convexity" in the discussion of Theorem 1. As discussed in Section 2.2, the objective function $L$ is always convex and satisfies a quasi strong convexity condition in (5). The discussion after Theorem 1 seems relevant to "conserving *strong* convexity" instead. Please improve the discussion about it. (P.S. On a related note, there is a small typo before (22), which wrongly stated that "$L$ is $\mu$-strongly convex".)

3. The distinction between long-range and short-range dependence was mentioned briefly in Sec. 3.1 and the paper proposes to distinguish them by checking if $\upsilon > 1/2$ or not. This choice of threshold is not clear from the context, and the reviewer guesses that it is related to the discussion in "Behavior for bias" after Theorem 1. If it is so for the latter case, then such choice should be justified as early as in Sec 3.1.

4. For the results about averaged stochastic streaming gradient, it is noted that the main result Theorem 2 is presented with $p=4$ instead of $p=2$ as in Theorem 1. Why is there a change of moment's order? Such has not been discussed in the paper.

**Strengths And Weaknesses:**

On the upside, I think the results are mostly correct (though I did not check every details in the appendix due to the time limit) and may be useful upon further polishing.

However, as a general comment, I find the paper very challenging to read due to the extensive use of constants/notations, which maybe even harder for general readers. The reviewer believes that the presentation of the main results in this paper should be substantially simplified before publication. Detailed comments and suggestions can be found in the "Requested Changes".

---

> ### Author Response · Authors · 2023-06-16
> **Response to review**
>
> Thank you for your positive feedback and your valuable suggestions, which have helped us
> improve the paper further. We also want to emphasize that we have performed a careful polishing of the paper, including
> refining the presentation, revising the notation, restructuring the content, and improving the discussions to improve
> clarity and readability. Here are the responses to Requested Changes:
> 1. The proof of Lemma 1 and the condition mentioned in Theorem 1: We appreciate the reviewer's observation regarding the condition presented in Lemma 1 and its apparent difference in Theorem 1. We apologize for the confusion caused. As $\nu_{t}$ is set equal to $n_{t}^{-\nu}$, and $n_{t}=\lceil C_{\rho}t^{\rho}\rceil$, then $\rho=0$ corresponds to $\nu_{t}$ being constant. We hope this makes it more clear. In addition, in the revised version we now present and discuss these function forms in order to avoid confusion.
> 2. Discussion on "conserving convexity" and "conserving strong convexity":
> We apologize for the confusion caused by the discussion of "conserving convexity" in relation to Theorem 1. You are correct that the objective function is always convex and satisfies a $\mu$-quasi-strong convexity condition. We have now added a discussion in Section 2.1 entitled "$F$ being $\mu$-quasi-strongly convex does not guarantee the $\mu$-quasi-strong convexity of $(f_{t})$". It is crucial to understand that while the objective function $F$ satisfies the $\mu$-quasi-strong convexity assumption, the individual loss functions $(f_{t})$ may not exhibit the same property. This distinction plays an essential role in our convergence analysis as it interacts with the time-dependency of the problem and the presence of biases. In section 3, we explore this relationship in detail and investigate its implications for the convergence behavior. Specifically, we examine how the level of dependence, biases, and the $\mu$-quasi-strong convexity conditions intertwine and affect the convergence properties of the optimization process. Additionally, we corrected the typo before equation (22) where it incorrectly states that "F is strongly convex."
> 3. Distinguishing long-range and short-range dependence: We appreciate the reviewer's feedback regarding the distinction between long-range and short-range dependence. This should be justified clearly in Section 2.2 now. We apologize for the lack of clarity. In the revised version, we provided a more explicit justification for the choice of threshold and its connection to the discussion on $\nu_{t}$.
> 4. Change of moment’s order in Theorem 2: The reviewer correctly points out the difference in the moment’s order
> between Theorem 1 and Theorem 2. We apologize for not discussing this change in the paper. The analysis for
> the averaged stochastic streaming gradient relies on the standard decomposition of the loss terms, which allows
> the Cramer-Rao term to emerge. To facilitate this analysis, it is standard to consider the fourth-order moments,
> requiring the assumptions to hold for $p=4$. This choice has been made in previous works as well (e.g., see Bach
> and Moulines (2011); Gadat and Panloup (2023); Godichon-Baggioni et al. (2023)). We clarified this point and
> provided a proper explanation in the revised version of the paper

---

### Review · Reviewer_BT8S · 2023-05-22

**Summary Of Contributions:**

This paper studies the stochastic gradient descent (SGD) for the streaming data where the time-dependent and biased estimate of the gradient is available. In particular, the authors derive the general form of convergence rates under mild assumptions for SGD and averaged SGD. Finally, the theory is verified with several experiments on synthetics dataset, AR model, ARCH model, and so on.

**Audience:**

Yes

**Claims And Evidence:**

Yes

**Requested Changes:**

See weaknesses for the detail.

**Strengths And Weaknesses:**

Strengths:

This theory builds on a mild assumption about the bias of gradient estimate (Assumption 2-p). To my knowledge, this assumption is considerably mild.

Weaknesses:

- The theory seems to be an extension of [Godichon-Baggioni et al. (2021)]. Thus, it would be nice to clarify the technical challenge from this previous work.
- I could not understand the following sentences for $\kappa$: from *``Obviously, $(\nu_t), (\kappa_t),$ and $(\sigma_t)$ may
be considered as uncertain terms depending on the streaming batches $(n_t)$.''* to *``Having, $\sigma,\kappa \in [0,1/2]$ follows directly from Godichon-Baggioni et al. (2021), since $\sigma=\kappa=1/2$ corresponds to the i.i.d. case.''* If $\kappa_t \sim n_t^{-1/2}$  following this description, then we can deduce $\nabla l_t(\theta)=\nabla l_t(\theta^*)$ in $L_p$ as $n_t \to \infty$  from Assumption 2-p. Clearly, this is false in general. Please correct me if my understanding is incorrect. (Moreover, I could not find explanations or specific examples of these constants in Godichon-Baggioni et al. (2021), although the authors cite this paper here.)
- The notation of $l_t$ is confusing. At first glance, I thought the objective $L(\theta)=\mathbb{E}[l_t(\theta)]$ itself is time-dependent because of this notation. Moreover, the mini-batch $l_t=(l_{t,1},\ldots,l_{t,n_t})$ is also confusing because of the confliction of notations.
- Assumptions 2-p and 3-p with p=1/2 imply ABC-condition studied in the following papers. In this sense, ABC-condition is a milder assumption. It would be better to discuss the relationship.

[1] Gower, Sebbouh, and Loizou. (2021a). Sgd for structured nonconvex functions: Learning rates, minibatching and interpolation.

[2] Khaled and Richtarik. (2020). Better theory for sgd in the nonconvex world.

- Lack of citations to relevant papers which show the statistical optimality of SGD (for least-squares regression). For instance, see below.

[3] Dieuleveut and Bach (2016). Nonparametric stochastic approximation with large step-sizes

[4] Dieuleveut, Flammarion, and Bach (2017). Harder, better, faster, stronger convergence rates for least-squares regression.

---

> ### Author Response · Authors · 2023-06-16
> **Response to review**
>
> We appreciate the reviewer's thorough assessment of our paper. We sincerely appreciate the reviewer's insightful comments and suggestions. These has helped us improving the clarity and quality of our paper. Regarding the strengths and weaknesses identified by the reviewer, we address them as follows:
> 1. The reviewer mentions that our theory appears to be an extension of Godichon-Baggioni et al. (2023) and
> requests clarification regarding the technical challenge from this previous work. We apologize for any confusion
> caused by the lack of clarity in our paper. In the revised version, we explicitly discussed the relationship with
> Godichon-Baggioni et al. (2023) and clarified the technical advancements and challenges presented in our work
> compared to theirs, e.g., see section 1 where we state our contributions.
> 2. We apologize for the confusion caused by the sentences from 'Obviously, and may be considered as uncertain terms depending on the streaming batches'. We acknowledge the incorrect reasoning presented in these sentences. In the revised version, we provided a detailed explanation that accurately reflects the deduction from the unbiased i.i.d. setting in Godichon-Baggioni et al. (2023). In section 3.1, we define and discuss the functional forms of the learning rate $(\gamma_{t})$, the uncertain terms $(\nu_{t})$ and $(\sigma_{t})$, and the streaming batch $(n_{t})$.
> 3. We understand the confusion caused by our notation, and we apologize for the inconvenience caused. In the revised version, we changed the notation to ensure clarity and avoid conflicts.
> 4. The reviewer points out that Assumptions 2-p and 3-p with p=1/2 imply the ABC-condition studied in several papers, suggesting that the ABC-condition may be a milder assumption. We appreciate this observation and included a discussion in the revised version regarding the relationship between our assumptions and the ABC-condition, along with references to the relevant papers mentioned  (Gower et al., 2019, 2021; Khaled and Richtárik, 2023). Additionally, we apologize for the lack of citations to relevant papers demonstrating the statistical optimality of SGD for least-squares regression. We thank the reviewer for bringing this to our attention. In the revised version, we could include citations to the papers mentioned  (Dieuleveut and Bach, 2016; Dieuleveut et al., 2017) to provide a more comprehensive and accurate reference list.

---

> > ### Comment · Reviewer_BT8S · 2023-07-05
> > **Response to the authors**
> >
> > Thank you for the response. My concerns have been well addressed. The quality of the paper has been improved by the revision, making the paper easier to read.

---

### Review · Reviewer_yQ8R · 2023-06-04

**Summary Of Contributions:**

The paper provides rates for stochastic gradient descent with time-dependent streaming data.

Authors consider a stochastic gradient descent scheme in which the stochastic oracles are not independent along the sequence. They provide convergence rates under classical assumptions on the function namely expected smooth and \mu quasi strong convexity, both in terms of expected squared distance to the optimum $E[||\theta_n-\theta_*||^2]$ and for a weighted Polyak-Ruppert average iterate.
The correlation between consecutive gradient oracles is illustrated on classical (arch, etc.) models.

The results leverage the analysis of SGD with conditionally biased stochastic gradients (Lemma 1), that are combined with assumptions on the conditional bias to obtain the main convergence theorems.

The application to the AR case is interesting and the discussions following theorem 1, on the choice of the size batches to obtain
convergence.



**Audience:**

Yes

**Broader Impact Concerns:**

-

**Claims And Evidence:**

Yes

**Requested Changes:**


### Writing:
The writing of the paper could be improved: the technical contributions are not completely clear and the are many typos (see below).

In my opinion, the paper would be easier to read if a Lemma, corresponding to eq. 9 was stated, or possibly Lemma 1 was stated immediately after Assumption 1-p to 3-p, with precise explanations on the proof technique and the novelty regarding theory of SA, if any.
Then in a Second subsection,  stating the specific choices of $\nu, \kappa, \sigma$ as _assumptions_, with a corollary.

The discussion paragraph following assumption 3-p should be reorganized:  the link between the size of the minibatch and  Assumption 1-p is critical for the rest. Making a formal statement would help.

---
### Related work
Several related references are missing or mentioned but the relationship should be clarified.

On Markovian dependent LSR:
Least Squares Regression with Markovian Data: Fundamental Limits and Algorithms https://proceedings.neurips.cc/paper_files/paper/2020/file/c22abfa379f38b5b0411bc11fa9bf92f-Paper.pdf

On SGD with bias.
https://arxiv.org/pdf/2008.00051.pdf

--> Litterature on clipping
https://arxiv.org/pdf/2005.10785.pdf and related refs

--> Litterature on biased compression
https://proceedings.neurips.cc/paper/2020/hash/ef9280fbc5317f17d480e4d4f61b3751-Abstract.html and related refs

---
### Typos

Footnote 1: Momentum and Nesterov accelerated methods are generally not considered accelerated.

The article needs to be fully proofread (including the proofs as there are still several typos). For example:
page 6:
-  v_t stands for \nu_t?
- allowS
- oF the  form
page 7:
- combining both more ideal

page 20:
- since L is $\mu$ strongly convex [such an assumption is not made]
- it's imply
- $(\kappa_t)$ and $(\gamma_t)$ is

**Strengths And Weaknesses:**

---
Strengths
Overall, the results are technically correct and can be of interest to members of the community, so publication in TMLR is well-suited. However, the paper should first be proofchecked and the writing can be improved.

---

###  Questions

#### Assumption 1-p
- what does the external expectation correspond to in assumption 1-p? If the assumption is valid for any $\mathcal F_{t-1}$ measurable variable, isn't it valid for any $\theta$?

####  On theorem 1:
- is there any technical novelty in the proof techniques?
- what do the authors mean by batches of size $t^\rho$: is it an upper bound? $n_t$ is  in  $N$. The second paragraph in Section 3.2 needs to be clarified. In the proof of Theorem 1, it seems that it is simply assumed that $n_t= C_\rho t^\rho$ for any t, and for some $\rho\in [0,1)$.
- I am not sure to understand why constants like $\mu_\nu =\mu -1_{\rho=0} D_\nu C_\rho^{-\nu}$ have to appear. The condition $\rho=0$ seems artificial, as I could consider $\rho=10^{-30}$ and that would seemingly change the rate. But again this is not very clear: I suspect that $C_\delta$ depends on $\rho$ which is probably the case from the derivation following eq. 23. If so, that dependency should be made clear in the theorems.
- In this derivation, I suspect that C_\delta is chosen as a constant "possibly smaller than 1" such that....  I fail to see how it can be ensured that $C_\delta$ would verify the condition and be larger than  1.

---

> ### Author Response · Authors · 2023-06-16
> **Response to review**
>
> We appreciate the reviewer's careful evaluation of our paper and their recognition of its contributions. We agree that proofreading and improving the writing are important steps, and we apologize for any confusion caused by the writing style and typos. In the revised version, we will thoroughly proofread the entire paper and make necessary improvements to enhance clarity and readability. We sincerely appreciate the reviewer's valuable feedback, which will help us enhance the clarity, technical correctness, and overall presentation of our paper. In response to the reviewer's specific questions and concerns:
>
> Assumption 1-p: We apologize for the lack of clarity in the current version of the paper. The external expectation
> in Assumption 1-p corresponds to the general form for mixing, e.g. see Bradley (2005); Rio (2017); Doukhan
> (2012). We will clarify this point in the revised version. Here, we have added a discussion in section 2.2. about
> this assumption, and how it can be verified using mixing coefficients (Rio, 2017).
>
> Theorem 1:
> - Regarding the technical novelty in the proof techniques of Theorem 1, we appreciate the reviewer’s question.
> Proofing techniques are not new in themselves, but the loss of the iid hypothesis introduces a bias, for
> instance, which, combined with the use of streaming settings, makes proofing particularly technical and
> difficult. The careful study of the interplay between temporal dependence and biases is novel. It requires
> new Assumption 1-p, and the adaptation of the proofs of the iid setting, given this new assumption. We have
> added this discussion to the paper.
> - Regarding the clarification of batches of size, we apologize for the lack of clarity. We will provide a clearer
> explanation of this choice in the revised version to avoid confusion. Indeed, in section 3.1, we clearly
> define the mini-batches in the following way: inspired by the work of Godichon-Baggioni et al. (2023), we adopt a formulation for time-varying mini-batches $(n_{t})$ given by $n_{t}=\lceil C_{\rho}t^{\rho} \rceil$, where $C_{\rho}\in\mathbb{N}$ and $\rho\in[0,1)$, ensuring that $n_{t}\in\mathbb{N}$. This formulation encompasses various scenarios, including classical (online) SGD methods for $C_{\rho}=1$ and $\rho=0$, (online) mini-batch SGD procedures with both constant and time-varying sizes when $C_{\rho}\in\mathbb{N}$ and $\rho=0$ or $\rho\in[0,1)$, respectively, as well as the Polyak-Ruppert average of (online) time-varying mini-batches.
> - Regarding the dependence penalized convexity constant $\mu_{\nu}$, we have added a discussion in section 2.1 entitled "$F$ being $\mu$-quasi-strongly convex does not guarantee the $\mu$-quasi-strong convexity of $(f_{t})$", together with follow-up discussion after Theorem 1 and 2. It is essential to grasp the distinction between the objective function $F$ satisfying the $\mu$-quasi-strong convexity assumption and the individual loss functions $(f_t)$ potentially lacking this property. This distinction is crucial in our convergence analysis as it interacts with the time-dependency of the problem and the presence of biases. Consequently, the introduction of the dependence-penalized convexity constant is driven by this constraint.
> - We apologize for the confusion regarding the appearance of constants, but these can be found in the appendix.
> To avoid excessive complexity in the theorem statement, we introduce these constants to encapsulate the
> negligible terms and focus on the main terms. However, for the sake of completeness, we provide a detailed
> explanation of all the terms in the appendix. Regarding $C_{\delta}$, we have added the verification to the appendix.
> Moreover, we can also refer to Godichon-Baggioni and Tarrago (2023) for more examples of this.
>
> Requested Changes:
> - Writing: We appreciate the reviewer’s suggestion to improve the writing by stating a Lemma corresponding to
> Equation (11) or immediately after Assumption 1-p to 3-p. We agree that providing a clear and formal statement
> of the lemma along with explanations on the proof technique and any novelty in the theory would enhance the
> readability of the paper. We have incorporated this suggestion and reorganized the subsections accordingly.
> - Related work: We thank the reviewer for pointing out missing references and suggesting clarification regarding
> the relationship with previous works. In the revised version, we included the references provided (Markovian
> dependent LSR, SGD with bias, literature on clipping, and literature on biased compression) and provide a
> clearer discussion of the relationships and connections between our work and the cited papers.
> - Typos: We apologize for the many typos and appreciate the reviewer’s diligent proofreading. In the revised
> version, we have proofread the entire paper, including the proofs, to correct the mentioned typos and enhance the
> overall quality.

---

> > ### Comment · Reviewer_yQ8R · 2023-06-27
> > **This is the response to Reviewer gnir**
> >
> > Dear authors,
> > It seems you duplicated twice the same response and have not replied to my review.
> >
> > Also, please update the revised version *with highlighted changes in a different color*. It is very difficult or impossible for us to track the changes and evaluate the revision without that.

---

> > > ### Author Response · Authors · 2023-06-27
> > > **Correct review is uploaded + a brief overview of the major (structural) changes made in the revised version**
> > >
> > > Dear reviewer,
> > >
> > > You are absolutely correct, and we sincerely apologize for the mistake of reporting the same response. We have now uploaded our response to your specific review by editing the incorrect submission.
> > >
> > > Regarding the revised version, we want to assure you that we have carefully reviewed and incorporated all the valuable comments and suggestions provided by the reviewers, including your feedback. We deeply appreciate the time and effort you have dedicated to reviewing our paper.
> > >
> > > We understand the importance of making it easy for reviewers to track the revisions and evaluate the changes made. We apologize for any confusion caused by not highlighting the modifications in a different color. Given the extensive nature of the revisions, it would be challenging to individually highlight all the changes. However, we have prepared a comprehensive summary of the major modifications made in the revised version. This summary aims to provide you with a clear overview of how your comments have influenced the paper and how we have addressed them in our revisions.
> > >
> > > Here is a brief overview of the major (structural) changes made in the revised version. We hope that this summary will assist you in evaluating the impact of your feedback on the revised manuscript:
> > >
> > > 0. Abstract: The abstract has been revised to clearly state our contributions.
> > > 1. Introduction: We have presented our contributions in a concise point form to provide readers with an easily understandable overview.
> > > 2. Problem Formulation, Assumptions, and Methods: The structure and order of this section have been modified as follows:
> > > - We now define the objective at the beginning of Section 2 (with changed notation according to Reviewer BT8S).
> > > - Our assumptions on objectives are presented in Section 2.1.
> > > - Assumptions on gradient estimates, along with extended discussions, are presented in Section 2.2.
> > > - Our stochastic streaming optimization methods are introduced in Section 2.3.
> > > 3. Convergence Analysis:
> > > - We have added Lemma 1, which derives a recursive relation for $\delta_{t}$ (suggested by Reviewer yQ8R).
> > > - Section 3.1 has been included to clearly define and discuss the learning rate $(\gamma_t)$, uncertainty terms $(\nu_t)$ and $(\sigma_t)$, and time-varying mini-batches $(n_t)$.
> > > - The discussions in Sections 3.2 and 3.3 have been revised to explicitly relate to our framework in Section 2.
> > >
> > > If there are any areas that remain unclear or if you have any further questions, please do not hesitate to reach out to us. We are more than happy to provide any additional clarification or information you may need.

---

> > > > ### Comment · Reviewer_yQ8R · 2023-07-07
> > > > **Follow up**
> > > >
> > > > I thank the authors for their response, however, I still do not understand several parts of the paper:
> > > > - my question was on the discontinuity wrt  $\rho$ of the main bound of the paper: this is  an obvious question when reading the bound, which it  seems is not commented in the paper:
> > > > to make it clearer: In practice, running the algorithm with a batch size of $n_t=10$ over $t\le 10^{20}$ iterations (which is more than any practical  purpose) could be written as
> > > > - $C_\rho =10, \rho=0$
> > > >
> > > > OR
> > > > - $C_\rho = 9,$ and $\rho=10^{-3}$
> > > >
> > > > However, the bound and even the condition for convergence is seemingly different. Obviously, again, this comes from the non-asymptotic bound in which one of the constant (either C_\delta or the speed of O) diverges with $\rho$ to 0, but should be made clear. Maybe this only  surprised me, but having a different concept of strong convexity and a discontinuity at rho =0 is very surprising and  seems to be an artefact of the proof.
> > > >
> > > > - I still do not understand the derivation between eqs 22 and 23. Can the authors clarify this paragaph
> > > >
> > > > - what does the external expectation correspond to in assumption 1-p? If the assumption is valid for any measurable variable, isn't it valid for any ?
> > > >
> > > > Although it is a minor point. I maintain my question: the assumption is made for all $\theta$  $\mathcal F_{t-1}$-measurable, thus should hold for all diracs at $theta_0$, for any  $theta_0$: I do not see what the external assumption is for.
> > > > [If providing references to any paper of more than 20 pages (and  even more considering books), a reviewer would  appreciate a more precise reference than the full book.]
> > > >
> > > > Many thanks in advance

---

> > > > > ### Author Response · Authors · 2023-07-11
> > > > > **Response to follow-up questions**
> > > > >
> > > > > Thank you for your follow-up comments. We sincerely appreciate your thorough assessment of our paper. Below, you will find our responses to your follow-up questions:
> > > > >
> > > > > Regarding your remark on the number of iterations, we agree that the number of iterations can be expressed in terms of the total sample size $N_T$. As you mentioned, for a given $N_T$, the number of iterations $T$ can be approximated as $T = \left( N_T (1+\rho)C_{\rho}^{-1} \right)^{\frac{1}{1+\rho}}$. Taking into account the example in Figure 1, where $N_T=10^5$, $C_{\rho}=8$ and $\rho=0.5$, we can observe that this allows for approximately $706$ steps, and the estimation of the gradients becomes more precise, leading to convergence. However, we agree that choosing excessively large values for $C_{\rho}$ or $\rho$ can result in less favorable outcomes in practice. For instance, if we take $C_{\rho}=64$ instead of $8$, we would have around $176$ iterations, as shown in Figure 1, and this may lead to less satisfactory results. Nevertheless, it is evident that our approach with varying (increasing) batch sizes outperforms constant batch sizes (and therefore SGD) in the non-i.i.d. case. The challenge lies in calibrating $\rho$ appropriately to achieve optimal performance in practice. The essence of choosing a small value for $C_\rho$ and $\rho$ in our analysis is to allow for as many gradient steps as possible while ensuring convergence. By selecting a small $C_\rho$ and $\rho$, we aim to strike a balance between taking frequent gradient steps for faster convergence and ensuring the stability and convergence of the optimization process. We will add a remark about this to the paper.
> > > > >
> > > > > For the derivations between Equation (22) and (23), we follow similar steps as outlined in previous works, such as Bach and Moulines (2011) and Godichon-Baggioni et al. (2023). We appreciate your observation and understand your concern regarding the different concept of strong convexity and the apparent discontinuity at $\rho=0$. We acknowledge that this discrepancy may appear surprising and could be perceived as an artifact of the proof. We acknowledge that the introduction of the indicator for $\nu$ is the reason for the lack of continuity with respect to $\rho$. Indeed, the introduction of the indicator for $\nu_t$ distinguishes between the cases of a constant $\nu_t$ and an decreasing $\nu_t$. When $\nu_t$ is constant, there is no hope to achieve a bound that approaches $\mu$ closely. On the other hand, when $\nu_t$ decreases, there is a possibility of obtaining a bound that can be as close as possible to $\mu$. In the context of the i.i.d. case, this choice corresponds to the optimal strong convexity parameter.
> > > > > To address the concern raised, we will discuss the presence of the indicator and the resulting discontinuity in the derivation may be considered an artifact of our proof in this specific section. We will also mention that there might exist a better bound by considering a "continuous" trade-off where we decrease $\nu_t$ in the bound while minimizing the other terms. This continuous trade-off could potentially provide a smoother transition between the cases of constant and decreasing $\nu_t$.
> > > > >
> > > > > Regarding assumption 1-p (now assumption 3-p): You are completely right, the extra conditions on the measurability of $\theta$ are superfluous and will be deleted. Regarding the verification of the mixing properties on Markov chains, we now refer readers to Chapter 9 in Rio (2017). This chapter provides comprehensive information on mixing coefficients and their relationships, including examples of linear, non-linear, and Markovian time series. Additional references such as Bradley (2005) and Doukhan (2012) offer further examples and insights into the mixing coefficients of weak dependence.

---

> > > > > > ### Author Response · Authors · 2023-07-11
> > > > > > **Response to follow-up questions + added remarks to the paper**
> > > > > >
> > > > > > We have included these comments in the paper on p. 8 and  p. 9, highlighted in blue.

---

> > > > > > > ### Comment · Reviewer_yQ8R · 2023-07-11
> > > > > > > **Thanks for your response - one last point**
> > > > > > >
> > > > > > > Thanks for taking into account my comments and for your response.
> > > > > > >
> > > > > > > One last point, my question in the last reply was on the derivation between eqs 23 and 24, and not 22 and 23, my apologies for the typo, especially on the choice of $C_\delta$ chosen larger than 1, but to be smaller than a quantity. + reference to conditions of Proposition 1 are unclear, etc.
> > > > > > >
> > > > > > > Can you rewrite that paragraph?

---

> > > > > > > > ### Author Response · Authors · 2023-07-12
> > > > > > > > **Response to last point**
> > > > > > > >
> > > > > > > > Thanks a lot. There were some troubles there which are now corrected.
> > > > > > > > I hope that this paragraph (whose modifications are highlighted in blue) is clearer now.

---

> > > > > > > > > ### Comment · Reviewer_yQ8R · 2023-07-12
> > > > > > > > > **ok**
> > > > > > > > >
> > > > > > > > > Thank you for the response. The quality of the paper has been improved by the revision, and most of my concerns addressed.

---

### Decision · Action_Editors · 2023-07-17

**Recommendation:** Accept as is

**Comment:**

The authors' engagement and incorporation of the reviewer's suggestions in their paper's presentation and writing have resulted in significant improvements. Given that the authors have addressed all the reviewer's suggestions, in my opinion, the paper is now ready for acceptance without further revisions.

**Audience:**

The paper offers an interesting analysis of Stochastic Gradient Descent (SGD) and its variants in a wide range of scenarios that have not been thoroughly explored in previous studies. This novel approach makes the paper relevant to a significant portion of the TMLR audience.

**Claims And Evidence:**

The main focus of this study is stochastic optimization within a streaming environment, where the gradient estimations are time-dependent and potentially biased. The authors undertake a non-asymptotic analysis of a general scheme that encompasses Stochastic Gradient Descent (SGD) and the associated Polyak-Ruppert averaging. The analysis yields interesting findings, suggesting that under certain assumptions and parameter selections, biased SGD methods have the potential to achieve performance levels similar to their unbiased counterparts. Additionally, the study demonstrates that the utilization of Polyak-Ruppert averaging can lead to faster convergence.

To validate their theoretical results, the authors conduct experiments using both simulated data and real-world time-dependent data, providing empirical evidence to support their findings.